# Scaling Inference-Time Computation via Opponent Simulation: Enabling Online Strategic Adaptation in Repeated Negotiation

**Xiangyu Liu** [1 2]  **Di Wang** [1]  **Zhe Feng** [1]  **Aranyak Mehta** [1]

## Abstract

While large language models (LLMs) have emerged as powerful decision-makers across a wide range of single-agent and stationary environments, fewer efforts have been devoted to settings where LLMs must engage in *repeated* and *strategic* interactions with unknown or dynamic opponents. In such settings, recipes built upon *offline* pre-training or fine-tuning, though robust against worst-case adversaries, do not fully exploit the capability of LLMs to adapt *online* based on interaction feedback. Instead, we explore the more natural perspective of scaling inference-time computation as a mechanism for adaptation, embedding the principles of a classical game-theoretical learning dynamic, *smooth Fictitious Play (sFP)*, into LLM inference: (i) for belief formation, we employ an auxiliary opponent model that in-context learns to imitate the time-averaged behavior of the opponent; (ii) for best response, we advance best-of-$N$ (BoN) sampling by simulating against the opponent model. Empirical evaluations on two distinct forms of repeated negotiation games demonstrate that our method enables significant performance improvement over online interaction compared to various baselines, offering a scalable and principled approach to repeated strategic decision-making without any parameter updates.

## 1. Introduction

Recent years have witnessed the remarkable success of large language models (LLMs) as central controllers across a broad spectrum of decision-making and reasoning tasks, including computer agents (Kim et al., 2023; Zhou et al., 2024b), robotics (Wang et al., 2024a; Cui et al., 2024), math-

/coding (Wei et al., 2022; Kojima et al., 2022; Jimenez et al., 2024). Notably, substantial research efforts have focused on developing effective policies for relatively stationary and single-agent decision-making environments (Hao et al., 2023; Yao et al., 2023).

Meanwhile, many applications also involve strategic interactions between the LLM-based agent and other decision-makers within the same system that are often unknown or may vary over time (Park et al., 2023; Zhang et al., 2024). One standard remedy involves computing static solutions such as the Minimax or Nash equilibrium through methods like *self-play*, exemplified by systems like AlphaGo (Silver et al., 2016; 2017) and recent strategic LLM agents powered by offline training (Bakhtin et al., 2022; Guan et al., 2024; Xu et al., 2025) or inference-time techniques (Kempinski et al., 2025; Light et al., 2025), which aim to converge to unexploitable policies against worst-case adversaries. However, such policies can be overly conservative especially in games involving both competition and cooperation (Leibo et al., 2017; Jaques et al., 2019) as shown later in Proposition 4.1. This highlights the necessity for LLM agents to adapt online to unknown or dynamic opponents and to progressively improve their decision-making policy by leveraging feedback accumulated during online interactions. Moreover, since online adaptation occurs dynamically at test time, recipes relying on gradient updates become less suitable, as they are data-hungry and introduce high latencies. Consequently, the paradigm of scaling *inference-time computation* emerges as the natural alternative, especially considering its recent success in single-agent domains like math reasoning (Jaech et al., 2024; Guo et al., 2025). This motivates our central research question:

*Can we enable online strategic adaptation for LLMs in repeated strategic decision-making by scaling inference-time computation?*

To answer this question, we focus on the natural language-based negotiation game, a widely adopted benchmark for evaluating LLMs' strategic capability (Lewis et al., 2017; Davidson et al., 2024; Bianchi et al., 2024; Xia et al., 2024b). These games present unique challenges for LLMs due to the necessity of reasoning over private information, mod-

[1]Google Research [2]University of Maryland, College Park. Work done while Xiangyu Liu is a research intern at Google Research. Correspondence to: Xiangyu Liu <xyliu999@umd.edu>.

*Proceedings of the 43$^{rd}$ International Conference on Machine Learning*, Seoul, South Korea. PMLR 306, 2026. Copyright 2026 by the author(s).

eling opponent behaviors, planning for long-term objectives, and engaging in strategic communication. More importantly, this setting offers an ideal middle ground: it is far more sophisticated than symbolic normal-form games (Akata et al., 2025; Kempinski et al., 2025), yet creates a more controlled environment than large-scale societies like Diplomacy (Bakhtin et al., 2022), allowing us to rigorously isolate the deep strategic reasoning required to adapt to a specific opponent from the confounding factors of general multi-agent group dynamics, enabling a precise analysis of inference-time scaling effects. We further introduce a new repeated setting, where agents must also leverage historical feedback to inform their actions over time. We propose scaling inference-time computation by embedding the principles of *smooth Fictitious Play* (sFP) (Brown, 1951; Robinson, 1951; Fudenberg & Levine, 1995) into practical LLM inference. Our approach explicitly allocates test-time compute to two decoupled FP modules: (i) **Belief formation**, where an auxiliary model in-context learns to mimic the opponent's *time-averaged* behavior from history; and (ii) **Best response**, where we advance best-of-$N$ (BoN) sampling by simulating full future trajectories for each candidate against the opponent model. By ranking strategies based on these computationally generated rollouts rather than static scoring, we effectively convert inference cost into strategic adaptation. We refer to our method as `BoN-oppo-simulation`.

**Contributions.** (1) We formalize and motivate our problem setting, demonstrating both theoretically and empirically the importance of engaging in repeated interactions and the failure of current LLMs to self-improve in such settings without additional inference-time interventions. (2) We then propose a general and principled framework for scaling inference-time computation to enable online strategic adaptation for repeated strategic decision-making. (3) Finally, we provide systematic empirical investigations, offering insights into the effectiveness of different candidate generation processes and evaluation strategies, as well as comparisons between thinking *wider* versus *deeper*, and demonstrate our framework achieves significant self-improvement.

## 2. Related Works

**Language models for negotiation games.** There has been a rich line of literature on negotiation games in various disciplines from game theory, economics, to psychology with a pre-defined symbolic action space. Beyond environments with standardized inputs and outputs, combining modern NLP and RL techniques for negotiation with unrestricted natural languages dates back to (Lewis et al., 2017), which trained an end-to-end recurrent neural network by imitating human dialogues followed by goal-based RL training and decoding. (He et al., 2018) further proposed to first generate the coarse dialogue acts and then use a generator to generate the actual natural dialogues. More recently, with LLMs as reliable natural language processing and understanding

interfaces, numerous works have attempted to benchmark the (native) negotiation ability in different negotiation settings (Davidson et al., 2024; Bianchi et al., 2024; Xia et al., 2024b). Meanwhile, there has also been a surging interest in improving the negotiation ability of LLMs with various techniques (Hua et al., 2024; Gemp et al., 2024; Liu et al., 2025; Zhang et al., 2025). These existing works mainly focus on how to learn a single policy with better performance in a single episode of the negotiation instead of enabling online adaptation and continual improvement over repeated interaction as in our paper.

**LLM agents for general strategic decision-making.** With LLMs being employed as the central controller for various (single-agent) decision-making problems (Yao et al., 2023; Shinn et al., 2023; Zhou et al., 2024a; Wang et al., 2024b), there have been efforts dedicated to evaluating the reasoning and decision-making capability of LLMs in the more challenging strategic environments including normal-form games (Akata et al., 2025; Brookins & DeBacker, 2024; Lorè & Heydari, 2023; Fan et al., 2024; Kempinski et al., 2025), bandits (Krishnamurthy et al., 2024; Nie et al., 2024; Xia et al., 2024a), expert problems (Park et al., 2025) with well-specified symbolic action space. There have also been related works on more specific game-theoretical domains, e.g., Diplomacy, Werewolf, as well as negotiation games above. These works can be roughly divided into two categories based on their methodology. The first line including (Bakhtin et al., 2022; Guan et al., 2024; Xu et al., 2024; 2025) leverages various training techniques (fine-tuning, self-play, RL, etc) aiming to learn a policy that can be deployed *statically* to outperform arbitrary adversaries. Such a static solution can be overly conservative and arbitrarily suboptimal in our repeated negotiation setting (cf. Proposition 4.1). Relying on parameter updates also makes it less suitable for online adaptation that occurs at test time. The second line including (Fu et al., 2023; Xu et al., 2023; Light et al., 2025; Yu et al., 2025; Kempinski et al., 2025) is free from parameter updates. These can be further divided into two sub-categories: (1) Input-level prompt engineering (Fu et al., 2023; Xu et al., 2023; Yu et al., 2025) (2) Output-level search (Kempinski et al., 2025; Light et al., 2025), Among these, only (Fu et al., 2023; Xu et al., 2023; Yu et al., 2025) are relevant to online adaptation, which we will further discuss and compare in Section 5 and Appendix D, while others still focus on the equilibria objective.

We refer more literature reviews on opponent modeling and inference-time scaling techniques in LLMs to Appendix B.

## 3. Preliminaries

The negotiation task has emerged as an important benchmark for examining the strategic reasoning abilities of LLMs. In this paper, we focus on two specific versions, the buyer-seller game and the resource exchange game (Rubinstein, 1982; Deng et al., 2024; Bianchi et al., 2024). Both

games involve an agent 1 and an agent 2 (i.e., LLMs).

- For the buyer-seller game, the buyer, who has a private maximum budget, aims to acquire an item from the seller who has a private production cost. If a deal is reached, the reward for the seller is defined as the difference between the deal price and the production cost, and the reward for the buyer is defined as the difference between the budget and the deal price. If no deal is reached, both get 0 reward.

- For the resource exchange game, each agent $i \in [2]$ holds a certain amount of different resources, for example, $n_i^X$ of $X$, and $n_i^Y$ of $Y$ with valuation of $v_i^X$ and $v_i^Y$ per unit of resource respectively for some $n_i^X, n_i^Y \in \mathbb{N}$ and $v_i^X, v_i^Y \in \mathbb{R}^{\geq 0}$. In such a setting, agents need to collaborate to trade less valuable resources for the more valuable ones. Each agent's reward is the net change in the total value of its resources.

In this paper, we are interested in the setting where the game is played repeatedly for $T \in \mathbb{N}$ episodes, where each episode further consists of (up to) a given horizon $H$ of turns (or steps). Formally, the protocol can be described as follows. We denote $x_1, x_2$ as the system prompts for describing the necessary game rules as well as the separate *private information* for the two agents. At each episode $t \in [T]$, step $h \in [H]$, agent $P(h) \in [2]$ takes an action $y_{P(h),h}^t = (y_{P(h),h}^{t,p}, y_{P(h),h}^{t,m})$, where $y_{P(h),h}^{t,p}$ encodes the structured information for a new proposal, acceptance, rejection, or waiting for a proposal, $y_{P(h),h}^{t,m}$ represents a free-format message to be sent to the opponent, and we define the space for $y_{P(h),h}^{t,p}$, $y_{P(h),h}^{t,m}$ as $\mathcal{Y}_{P(h)}^p$, $\mathcal{Y}_{P(h)}^m$ respectively. If agent 1 starts first, we have $P(h) = 2 - (h\%2)$; otherwise, $P(h) = 1 + (h\%2)$. We also let $\tau_h^t := (y_{P(1),1}^t, y_{P(2),2}^t, y_{P(3),3}^t, \cdots, y_{P(h-1),h-1}^t)$ denote the concatenated conversation history up to step $h$ within episode $t$, and $\mathcal{C}^{t-1} := (\tau_{H+1}^1, \tau_{H+1}^2, \cdots, \tau_{H+1}^{t-1})$ denotes the history of completed negotiations from episode 1 to $t - 1$, which serves as the context. At the end of episode $t$, agents 1 and 2 receive rewards $r_1^t$ and $r_2^t$, respectively. The game ends immediately if a proposal is accepted or rejected, or if the maximum number of turns is exceeded. By default, each agent $i \in [2]$ uses a policy in the form of $\pi_{i,h}^t(\cdot \mid \tau_h^t; \mathcal{C}^{t-1}, x_i)$ for each $h \in [H]$ where $P(h) = i$ and we denote the corresponding policy class as $\Pi_i^t$. Finally, we denote the expected reward of a single episode as $J_i(\pi_1^t, \pi_2^t) := \mathbb{E}[r_i \mid \tau_{H+1}^t \sim (\pi_1^t, \pi_2^t)]$. Throughout our paper, we mainly take the perspective of agent 1 and regard agent 2 as the opponent.

# 4. Methods
## 4.1. On the necessity of online adaptation
**There is no single dominant strategy.** Before resorting to online adaptation, one might ask: can we simply find a single offline strategy (e.g., via RL) that is optimal against all possible opponents? We formally show that such a domi-

nant strategy does not exist in our negotiation games

**Proposition 4.1.** *For both of our negotiation games, in a single episode, there does not exist a policy $\pi_1^\star \in \Pi_1$ such that for any $\pi_2 \in \Pi_2$, it holds $J_1(\pi_1^\star, \pi_2) = \max_{\pi_1 \in \Pi_1} J_1(\pi_1, \pi_2)$. In fact, for any $\pi_1^\star \in \Pi_1$, there exists $\pi_2 \in \Pi_2$ such that $J_1(\pi_1^\star, \pi_2) \leq \frac{\max_{\pi_1 \in \Pi_1} J_1(\pi_1, \pi_2)}{|\mathcal{Y}_1^m|}$, where we have omitted the episode index $t$ since there is only one episode and recall $\mathcal{Y}_1^m$ is the free-format negotiation message space of agent 1.*

We also empirically validate this non-dominance as follows. While prior work (Bianchi et al., 2024) identifies heuristic prompts like "cunning" or "desperate" as effective, we demonstrate such success is highly opponent-dependent. We expanded the prompt space using GPT-4o to obtain more diverse personas such as "fully rational", "fairness valuing", "emotionally reactive", and "Tit-for-Tat". Additionally, we include a brainstorming prompt, which asks the LLM to devise several strategies and evaluates them by itself. The specific prompts can be found in Appendix A.1. We report the pairwise performance of all seven prompts in Figure 1, revealing that no single prompt is consistently optimal.

**LLMs may fail to adapt (even when asked to).** Given the necessity of online adaptation through repeated interactions, we additionally examine whether LLMs can adapt naturally by simply conditioning on the negotiation history from past episodes. In the buyer-seller game, we let two Gemini-2.5-Flash models interact for 20 episodes and report the correlation between agent 1's average rewards of the first 5 episodes and the last 5 episodes in Figure 2, where we can see that most of the time, the performance remains stagnant.

## 4.2. Fictitious play for adaptive decision-making
Learning in games offers a solid theoretical foundation for equipping agents with adaptive decision-making capabilities when facing unknown or even adversarial opponents. One notable learning dynamic is *(smooth) Fictitious Play* (sFP) (Brown, 1951; Robinson, 1951; Fudenberg & Levine, 1995), where the agent maintains a belief over the opponent's actions and best responds to the belief at each episode. Specifically, taking the example of normal-form games, at each episode $t \in [T]$, the learning process for agent 1 can be described as follows

- **Step 1: Belief formation.** Agent 1 forms a belief about its opponent's policy $\widehat{\pi}_2^t \in \Delta(\mathcal{B})$ by tracking the empirical frequency of the opponent's historical actions. For each opponent's action $b \in \mathcal{B}$, if agent 2 has played action $b$ for a total of $k$ times over the past $t - 1$ episodes, the belief is $\widehat{\pi}_2^t(b) = k/(t-1)$, where $\mathcal{B}$ denotes the action space of agent 2.

- **Step 2: (Perturbed) best response.** Agent 1 computes a (perturbed) best response $\pi_1^t \in \Delta(\mathcal{A})$ against this belief $\widehat{\pi}_2^t$ such that for any $a \in \mathcal{A}$,

$$\pi_1^t(a) = \mathbb{P}\big(a \in \underset{a' \in \mathcal{A}}{\mathrm{argmax}} \, \mathbb{E}_{b \sim \widehat{\pi}_2^t}[r_1(a', b)] + \eta_t \epsilon(a')\big),$$

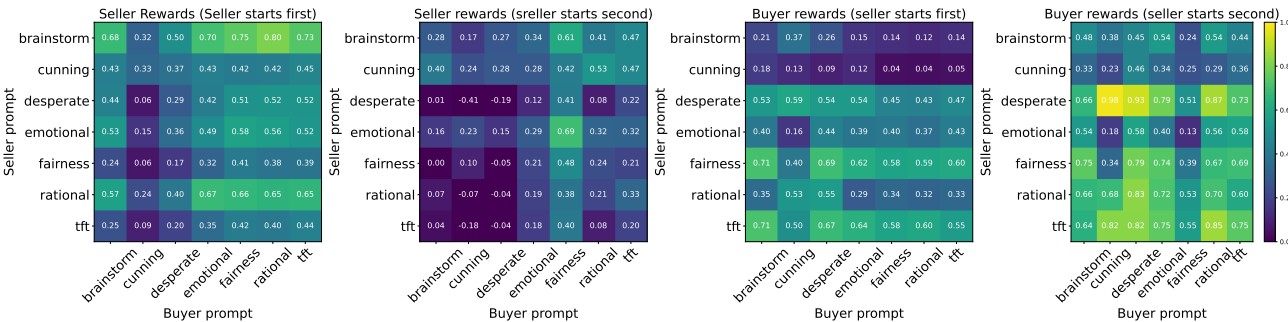

*Figure 1.* The pairwise normalized rewards among the 7 kinds of prompts for the buyer-seller negotiation games. Results shown for both buyers and sellers for both starting first and starting second.

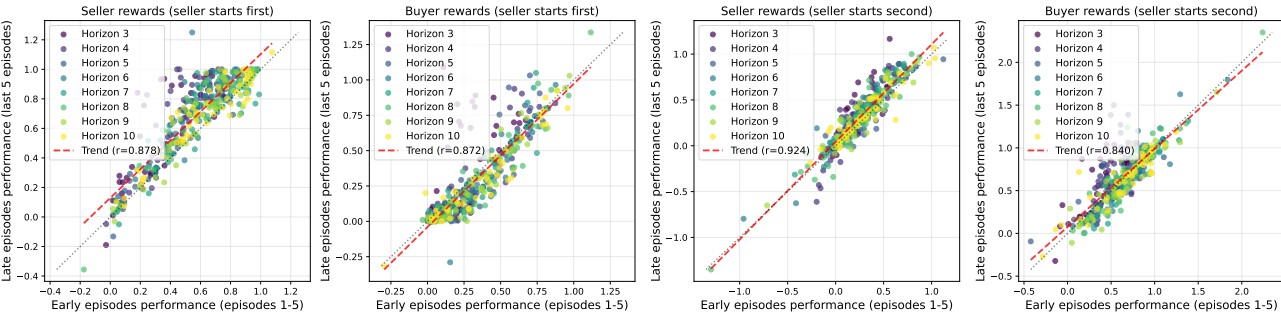

*Figure 2.* Correlation between the average normalized reward in the first 5 episodes and the last 5 episodes for buyer-seller negotiation games. Results are shown for all $7 \times 7$ different prompt pairs.

where $\mathcal{A}$ and $r_1 \in [0,1]$ denote the action space and reward function of agent 1. The perturbation term $\epsilon \in \mathbb{R}^{|\mathcal{A}|}$ is sampled from some given noise distribution $P_{\text{noise}}$ and $\eta_t \in \mathbb{R}^+$. Notably, it introduces randomness to agent 1's policy, preventing it from being exploited by the opponents, and is the key to achieving strong adaptive decision-making ability in the form of being *no-regret*.

**Proposition 4.2.** *Define the (external) regret as $Regret(T) = \max_{\pi_1 \in \Delta(\mathcal{A})} \sum_{t=1}^{T} V_1(\pi_1, \pi_2^t) - V_1(\pi_1^t, \pi_2^t)$, where we denote $V_1(\pi_1, \pi_2) := \mathbb{E}_{a \sim \pi_1, b \sim \pi_2} r_1(a, b)$ for any $\pi_1 \in \Delta(\mathcal{A}), \pi_2 \in \Delta(\mathcal{B})$. Suppose the perturbation is drawn from a standard Gaussian distribution. Then if $\eta^t = \Theta(1/\sqrt{t})$, it holds that $\mathbb{E}[Regret(T)] = \mathcal{O}(\sqrt{T})$ for any unknown policies $\pi_2^{1:T}$ played by the opponent.*

*Remark* 4.3 (Connections to online adaptation). Such guarantees are made possible by the equivalence between the sFP and the well-known online learning algorithm, follow-the-perturbed-leader (FTPL) (Kalai & Vempala, 2005), where the noise distribution can also be the Laplace distribution, Gumbel distribution, etc. (Abernethy et al., 2014). It implies that when $T$ becomes sufficiently large, the average performance of the agent 1 is comparable to that of the best policy in hindsight. In particular, when the opponent is stationary, as $T$ increases, the average performance of the agent 1 gradually approaches the optimum.

While this dynamic is elegant for normal-form games, implementing it directly in LLMs faces two fundamental computational barriers: (i) Exponentially large natural language

action space implies exact historical actions rarely repeat, making frequency-based belief formation impossible; (ii) The exact $\arg\max$ is intractable to compute over natural languages. In the following, we will discuss how to approximate these two steps by scaling inference-time computation.

### 4.3. Step 1: in-context opponent modeling

Translating **Step 1** to the language domain can be done by maintaining the *time-averaged* opponent's policy given the historic context. Specifically, an ideal solution to address the intractability of exponentially large natural language space would be leveraging the inductive bias of a pre-trained language model $\pi_\theta$ by fine-tuning it towards mimicking the opponent's behavior given the historical contexts $\mathcal{C}^{t-1} = (\tau_H^1, \tau_H^2, \cdots, \tau_H^{t-1})$ at each episode $t \in [T]$ using the objective of $\arg\max_\theta \sum_{t'=1}^{t-1} \sum_{h:P(h)=2} \log \pi_\theta(y_{2,h}^{t'} \mid \tau_h^{t'})$.

However, repeated fine-tuning is data-hungry and incur prohibitive overheads, making it less suitable for real-time online adaptation. Consequently, we propose to leverage an off-the-shelf LLM $\pi_2^{\text{oppo}}$ to *in-context learn* to imitate the behavior of the opponent using historical interactions $\mathcal{C}^{t-1}$. Specifically, at each episode $t \in [T]$ and step $h \in [H]$, where $P(h) = 2$, the opponent model $\pi_2^{\text{oppo}}$ takes the input of historical interactions $\mathcal{C}^{t-1}$, the current partial trajectory $\tau_{h-1}^t$ as well as the additional prompt $p$ that instructs $\pi_2^{\text{oppo}}$ to role-play the actual opponent to predict its behavior at this time step. This instruction prompt $p$ incorporates two key designs: (i) **Strategic summarization:** $\pi_2^{\text{oppo}}$ is required to

first explicitly reflect on the contexts $\mathcal{C}^{t-1}$ and summarize the high-level strategic behavioral patterns of the actual opponent; (ii) **Optimism:** We embed the principle of optimism in face of uncertainty (OFU), a principled exploration mechanism from online RL. Specifically, when $\pi_2^{\text{oppo}}$ is uncertain about how the actual opponent would have responded at the current step, it biased to predict outcomes that favor agent 1. We refer the specific prompts to Appendix A. Finally, we remark that **Step 1** of sFP (and our corresponding opponent modeling approach) maintains only the *time-averaged behavior* of the opponent, effectively treating the opponent as if it were *stationary*. However, this does not hinder the learner's ability of online adaptation when the opponent follows a time-varying policy sequence, as established in Proposition 4.2.

### 4.4. Step 2: BoN with opponent simulation

Implementing **Step 2** requires solving the intractable maximization problem over the natural language space. This is further complicated by the multi-turn nature of negotiation, where intermediate reward signals are missing. To address these computational hurdles, given the base LLM $\pi_1^{\text{base}}$, at each decision point of agent 1, $\tau_h$, where $P(h) = 1$, we first sample $N$ candidate actions $\mathcal{D}_{1,h} := \{y_{1,h}^1, \cdots, y_{1,h}^N\}$ from $\pi_1^{\text{base}}$. Different from the vanilla version of BoN which typically samples candidates i.i.d., we propose to first generate $N$ strategic proposals and then devise separate actions based on each proposal. We refer to this structured method as strategic brainstorming. Intuitively, this structured process ensures broader exploration of the strategy space. Crucially, during generation at episode $t \in [T]$ and each step $h \in [H]$, $\pi_1^{\text{base}}$ maintains not only (partial) history of the current episode, but also the history from episode $1$ to $t-1$. We explicitly allocate tokens to summarize and reflect on the history and then make corresponding decisions. Such summarization (Krishnamurthy et al., 2024) and reflection (Shinn et al., 2023) techniques have been shown to be necessary for enabling feedback-driven learning.

Now we evaluate each $y_{1,h}^k$ for $k \in [N]$ as follows. Due to the lack of an immediate reward signal, we propose to first follow $y_{1,h}^k$ at the current time step $h$, and *simulate* the entire future trajectory by following agent 1's base policy $\pi_1^{\text{base}}$ together with the opponent model $\pi_2^{\text{oppo}}$ to obtain the reward $\widehat{r}_1^k$ for agent 1. Then the algorithm picks the best candidate action $y_{1,h}^{k^\star}$ with $k^\star \in \arg\max_{k \in [N]} \widehat{r}_1^k$ and the decision-making process proceeds to the next time step. Finally, since both the candidate generation and opponent simulation involve inherent stochasticity, we empirically find that there is no need to further perturb the simulated reward as in **Step 2** of Section 4.2. To theoretically validate that better opponent model translates to more reliable evaluations and superior policy, we analyze the error propagation as follows.

**Theorem 4.4.** *Fix a given episode $t \in [T]$ with given initial*

*prompts $x_1$, $x_2$, historical context $\mathcal{C}^{t-1}$, opponent policy $\pi_2^t \in \Pi_2^t$, as well as the opponent model $\pi_2^{\text{oppo}}$. For any step $h \in [H]$ with $P(h) = 2$, we define the opponent error as $d_{TV}\left(\pi_{2,h}^t(\cdot \mid \tau_h^t; \mathcal{C}^{t-1}, x_2), \pi_{2,h}^{\text{oppo}}(\cdot \mid \tau_h^t; \mathcal{C}^{t-1})\right) \leq \epsilon_h$ for any decision point $\tau_h^t \in \mathcal{T}_h^t$. Then it holds for any policy $\pi_1^t \in \Pi_1^t$, step $h \in [H]$ with $P(h) = 1$, and $\tau_h^t \in \mathcal{T}_h^t$:*

$$\left| V_{1,h}^{\pi_1^t, \pi_2^{\text{oppo}}}(\tau_h^t) - V_{1,h}^{\pi_1^t, \pi_2^t}(\tau_h^t) \right| \leq \sum_{d=0}^{\lfloor (H-h-1)/2 \rfloor} \epsilon_{h+2d+1},$$

*where $d_{TV}$ denotes the total variation distance and the value functions are defined as $V_{1,h}^{\pi_1^t, \pi_2^{\text{oppo}}}(\tau_h^t) := \mathbb{E}_{\pi_1^t, \pi_2^{\text{oppo}}}[r_1 \mid \tau_h^t]$, $V_{1,h}^{\pi_1^t, \pi_2^t}(\tau_h^t) := \mathbb{E}_{\pi_1^t, \pi_2^t}[r_1 \mid \tau_h^t]$, and we assume the reward is properly normalized into range $[0, 1]$. Furthermore, let $\widehat{\pi}_1^t \in \arg\max_{\pi_1^t \in \Pi_i^t} J_1(\pi_1^t, \pi_2^{\text{oppo}})$ be the optimal policy against the opponent model. It holds that $J_1(\widehat{\pi}_1^t, \pi_2^t) \geq \max_{\pi_1^t \in \Pi_1^t} J_1(\pi_1^t, \pi_2^t) - \sum_{h \in [H]: P(h)=2} \epsilon_h$.*

This demonstrates that the evaluation errors and the optimality gap only scales *linearly* w.r.t. the model errors at each time step in our setting.

**A viewpoint of inference-time RL and extensions to higher-order BoN.** In principle, our framework creates a feedback loop equivalent to *one iteration* of the classical Policy Iteration (PI) algorithm, utilizing only inference-time computation. For each decision point $\tau_h$, where $P(h) = 1$ and candidate $y_{1,h}^k$, the simulation step functions as policy evaluation, where the simulated reward $\widehat{r}^k$ is in fact approximating $Q_{1,h}^{\pi_1^{\text{base}}, \pi_2^{\text{oppo}}}\left(\tau_h, y_{1,h}^k\right) := \mathbb{E}^{\pi_1^{\text{base}}, \pi_2^{\text{oppo}}}[r_1 \mid \tau_h, y_{1,h}]$. Subsequently, the ranking step constitutes policy improvement, where the new BoN policy chooses the optimal action as $\pi_1^{\text{BoN}}(\tau_h) := \arg\max_{y_{1,h} \in \mathcal{D}_{1,h}} Q_{1,h}^{\pi_1^{\text{base}}, \pi_2^{\text{oppo}}}(\tau_h, y_{1,h})$, constructing an improved policy $\pi_1^{\text{BoN}}$ from the weaker base policy $\pi_1^{\text{base}}$. This perspective reveals a new axis for scaling inference compute, distinct from increasing sample size $N$ or utilizing an auxiliary opponent model for simulation: one can repeatedly *sharpen* the base policy by $\pi_1^{(l)} \xleftarrow{\text{BoN-oppo-simulation}} \pi_1^{(l-1)}$ for $l = 1, 2, \cdots$, where $\pi_1^0 := \pi_1^{\text{base}}$. By the guarantee of PI, this will finally converge to the best response against the $\pi_2^{\text{oppo}}$ but without updating the parameters of $\pi_1^{\text{base}}$. We remark that recursively applying this operator is also conceptually similar to Monte-Carlo Tree-Search (MCTS). Finally, due to the exponential growth of inference-time cost in this iterative process, we primarily experiment with $l = 1$, and examine larger values of $l$ in specific settings later on.

Finally, we refer an overall demonstration of our framework to Figure 7 and Algorithm 1.

### 4.5. Can our framework be implemented in just one LLM query?

It is in fact intriguing to ask whether our multi-component framework above can be *integrated into just a single but*

*potentially much longer LLM inference query?* To understand this, we design a specialized prompt to teach the base LLM to reason by combining all components of our framework (cf. the prompt template to Appendix A.5). At each time step, it will brainstorm $N$ high-level strategies, devise concrete actions, simulate what would happen if it follows each candidate, and finally returns the simulated rewards to pick the best candidate. Note the key difference compared with our framework above is that the long simulation traces now happen purely in the LLM's native thinking/CoT. We call this *BoN with CoT simulation*. This does not only serves as an interesting baseline but also help us understand whether the *default thinking ability* of large reasoning models adopted by training heavily on inherently single-agent tasks like math and coding suffices for strategic reasoning.

## 5. Experimental Results

**Experimental setups.** We let our algorithm or baseline methods operate as one agent powered by an LLM to compete with the opponent also powered by an LLM. As noted by (Xia et al., 2024b; Bianchi et al., 2024), in such negotiation games, both the role (seller vs. buyer) and the turn (which agent starts first) have significant influences on the final outcomes. Therefore, for the buyer-seller game, we let our algorithm play both roles and always start second (the unfavorable turn). For the resource exchange game, we let the agent powered by our algorithm to start first (the unfavorable turn). For specifications of negotiation environments, we mainly follow (Bianchi et al., 2024): we set the seller's production cost as 43, buyer's budget as 63[1]. For the resource exchange game, we set $n_1^X = n_2^Y = 25$, $n_1^Y = n_2^X = 5$, $v_1^X = v_2^Y = 0.5$, $v_1^Y = v_2^X = 2.5$. The default horizon $H$ of one episode is 10.

We compare against a comprehensive suite of methods that also require *no parameter updates*: **(i) Standard inference:** *Baseline* (zero-shot), *Baseline w. Thinking* **(ii) Inference-time scaling:** *BoN-eval* (BoN with an evaluation model), *BoN-simulation* (cf. Section 4.5), and *BoN-oppo (iid)*, where candidates are sampled without structured generation. **(iii) External adaptive methods (cf. Appendix D for more introductions):** Approaches from (Fu et al., 2023) (AI Feedback), (Xu et al., 2023) (experience reflection), and (Yu et al., 2025) (private information prediction). Our method is denoted by the shorthand *BoN-oppo*. Unless otherwise stated, we set $N = 5$. For the opponent model or evaluation model, we use the same base LLM as the acting agent, with one exception: when both the acting agent and opponent are powered by Gemini-2.5-Flash, we instead use Gemini-2.5-Flash-Lite for opponent modeling to avoid self-modeling bias. All results are averaged over 10 random runs.

**Scaling inference compute enables robust adaptation across diverse opponent dynamics.** All opponents in our setting are inherently dynamic with base model powered by Gemini-2.5-Flash, maintaining full negotiation history and evolving their behavior across episodes based on contexts. We evaluate performance across a spectrum of opponent sophistication. **(1) Performance against general-purpose dynamic agents.** We first evaluate against standard LLM agents that adapt naturally via context accumulation. Figure 3a and Figure 3b illustrate the learning curves, where our method achieve the most significant and consistent performance gains compared to baselines. We also validate this across diverse model families (Claude-Sonnet-4, Qwen3, Llama-3.3) in Table 1, demonstrating that allocating compute to explicit opponent simulation unlocks strategic capabilities that other methods may fail to elicit. Notably, *BoN-simulation* often stands as the second best while and *Baseline w. Thinking* can even underperform the naive baseline, revealing that simply increasing "thinking time" is insufficient for strategic reasoning. **(2) Performance against specialized adaptive agents.** To push the limits of adaptation, we compete against opponents powered by state-of-the-art methods (Fu et al., 2023; Xu et al., 2023; Yu et al., 2025). As reported in Figure 6, our approach consistently outperforms these methods. This confirms that our method by scaling computation on the output-level (together with necessary input-level prompting techniques introduced before) yields superior adaptation compared to relying solely on input-level prompting. **(3) Robustness to environmental stochasticity.** We also introduce non-stationarity into the environment itself by randomizing the opponent's private constraints (budget/cost) at every episode. As shown in Figure 4a, our method maintains robust performance, proving it adapts to the opponent's *behavioral policy* rather than merely memorizing static values. **(4) Opponent architecture generalization.** We further evaluate generalization by varying the opponent's backbone LLM beyond Gemini, reporting the results in Table 3. **(5) Social welfare evaluation.** For the resource exchange game involving more cooperation than competition, another important metric is social welfare, i.e., the sum of both agents' value of their respective resources after exchange. Figure 17 compares the social welfare achieved by pairs of baseline and BoN agents. We find that the highest social welfare is achieved when both agents rely on our method, suggesting that inference-time scaling can promote more efficient equilibrium outcomes. We refer example outputs of our agents to Appendix F.

**Mechanism analysis** We here provide detailed analysis on two important algorithmic ideas of our framework. **(i) Opponent modeling:** To evaluate whether the opponent model can provide more and more accurate simulation through the accumulation of the negotiation history, we compare the best candidate ranked by the simulation results

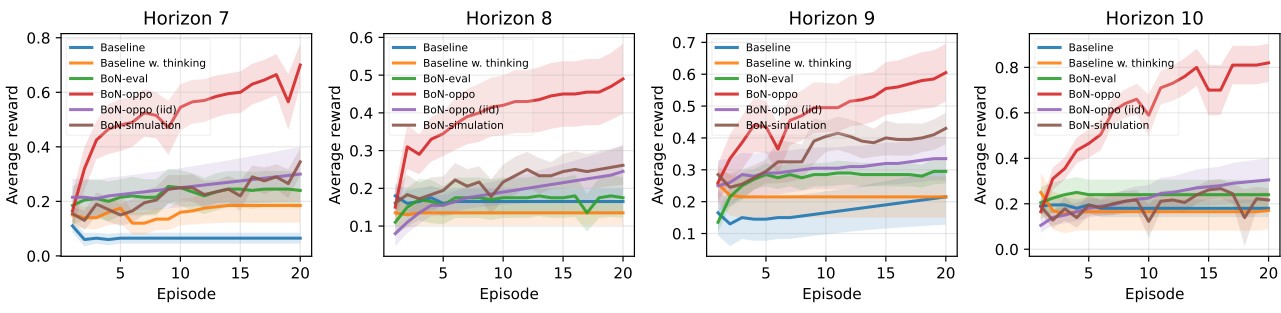

*(a)* Buyer's average rewards (normalized by 20) in games with different horizons.

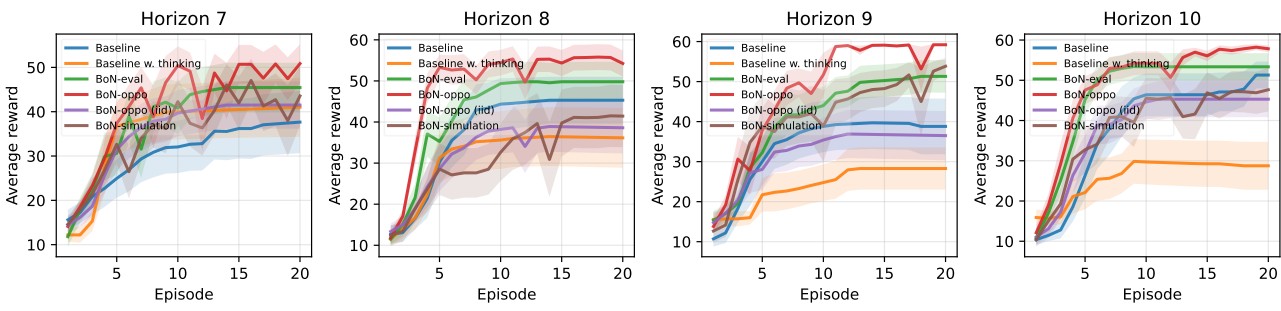

*(b)* Results for the resource exchange game.

*Figure 3.* Comparison of our method (red line) with 5 baselines introduced in Section 5.

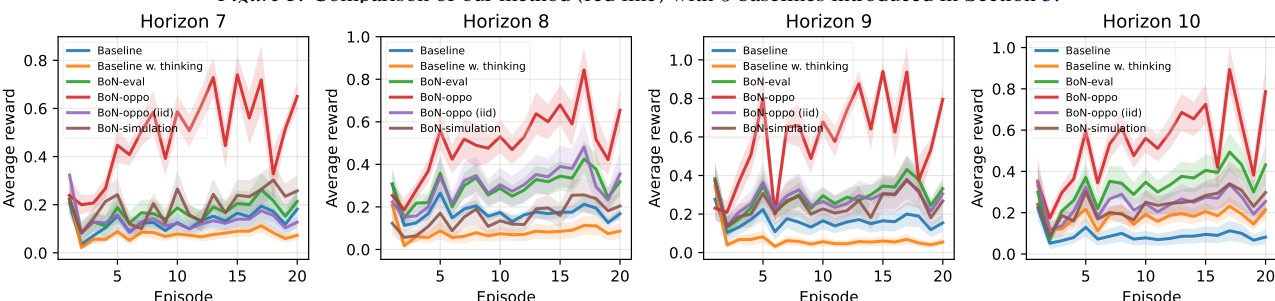

*(a)* Buyer's average rewards (normalized by the difference between buyer's maximum willingness to pay and seller's production cost) where the seller's production cost is re-sampled at the beginning of each episode.

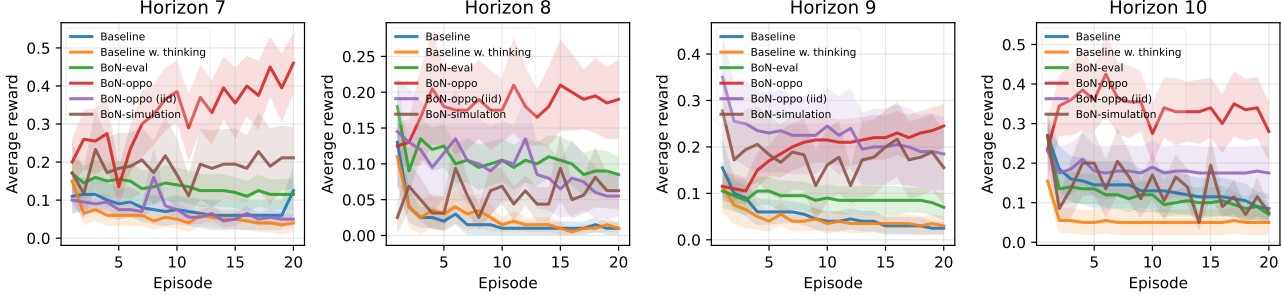

*(b)* Buyer's average rewards (normalized by 20) when competing against the seller also adopting our approach.

*Figure 4.* Comparison of buyer's performance under two seller behavior settings.

from the opponent model and the actual *oracle* opponent. The accuracy of different methods is reported in Figure 11, Figure 12, where the opponent model does provide increasingly more accurate simulation outcomes. **(ii) Strategic brainstorming:** Apart from the opponent model, another

major factor that affects the performance of BoN algorithms is the diversity of the candidates. One innovation of our algorithm comes from the structured generation process of brainstorming. We report semantic diversity of the candidate messages generated via strategic brainstorming and

| Model | Method | Buyer-seller game | | Resource exchange game | |
|---|---|---|---|---|---|
| | | Buyer | Seller | Starts first | Starts second |
| **Claude** | Baseline w. thinking | +2.02 ± 1.39 | -1.47 ± 2.05 | +27.45 ± 6.18 | -0.71 ± 1.16 |
| | BoN-eval | +0.68 ± 1.56 | +2.06 ± 1.75 | +20.69 ± 7.19 | +1.56 ± 0.40 |
| | BoN-simulation | +0.04 ± 2.05 | +1.36 ± 1.54 | +16.76 ± 6.78 | -11.15 ± 6.05 |
| | BoN-oppo (iid) | -1.16 ± 0.98 | +2.78 ± 1.67 | +28.00 ± 6.01 | +0.19 ± 0.56 |
| | **BoN-oppo** | **+3.02 ± 1.51** | **+2.80 ± 2.06** | **+30.65 ± 6.34** | **+1.92 ± 0.68** |
| **Qwen** | Baseline w. thinking | -0.42 ± 0.94 | +11.18 ± 2.00 | -7.17 ± 5.15 | +16.89 ± 4.80 |
| | BoN-eval | +4.06 ± 1.62 | +18.10 ± 1.97 | -0.05 ± 5.94 | +24.66 ± 2.85 |
| | BoN-simulation | +2.58 ± 1.65 | +11.12 ± 2.01 | -4.54 ± 5.91 | +9.17 ± 5.33 |
| | BoN-oppo (iid) | +1.60 ± 1.49 | +11.62 ± 1.93 | +1.95 ± 7.59 | +26.14 ± 1.16 |
| | **BoN-oppo** | **+10.04 ± 2.03** | **+18.54 ± 2.46** | **+8.95 ± 6.23** | **+29.65 ± 0.33** |
| **Llama** | Baseline w. thinking | — | — | — | — |
| | BoN-eval | -1.82 ± 1.88 | +10.64 ± 2.44 | -3.84 ± 6.41 | +9.55 ± 5.88 |
| | BoN-simulation | +0.30 ± 2.47 | +5.28 ± 3.11 | -16.26 ± 5.10 | **+10.34 ± 4.96** |
| | BoN-oppo (iid) | -1.78 ± 1.62 | -1.28 ± 2.44 | +5.95 ± 4.57 | +7.92 ± 5.62 |
| | **BoN-oppo** | **+4.80 ± 1.68** | **+14.74 ± 3.13** | **+13.27 ± 4.84** | +6.08 ± 5.83 |

*Table 1.* Performance *boost* of different inference-time methods over *Baseline* for three additional models. Results for our method (*BoN-oppo*) are shaded. Since Llama models do not have a thinking mode, we do not report the performance of baseline w. thinking.

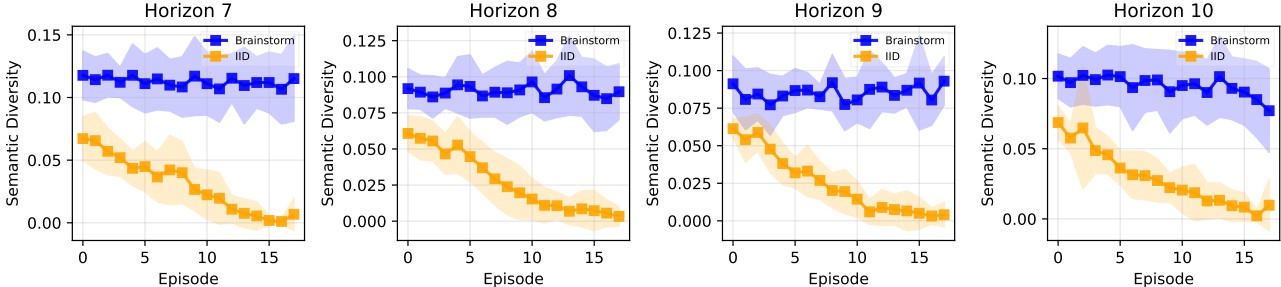

*Figure 5.* Semantic diversity of buyer's candidate messages, where the semantic diversity is calculated by the one minus of the average pairwise cosine similarity between the embeddings of candidate responses generated by the LLM.

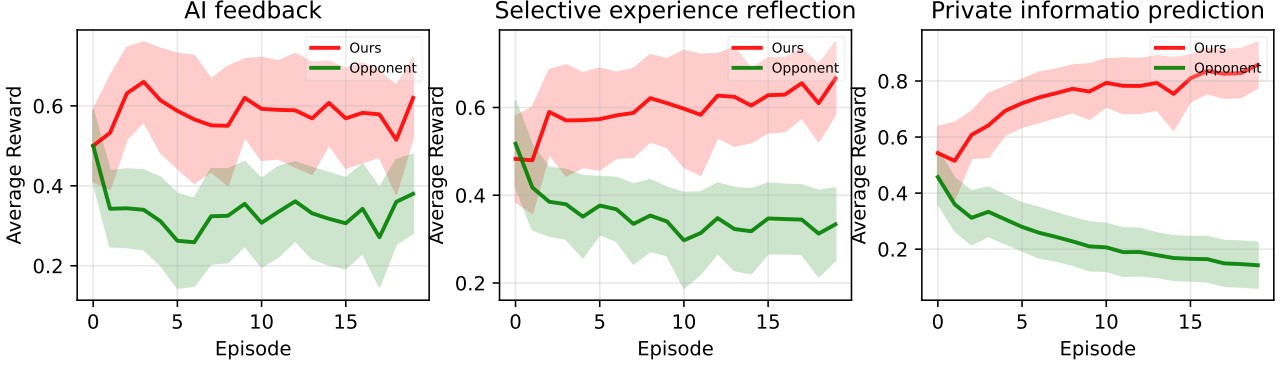

*Figure 6.* Normalized rewards of our approach competing against three kinds of strongly adaptive opponents powered by different kind of learning techniques, using AI feedback, using selective experience reflection, and using private information prediction.

i.i.d. sampling in Figure 5 and Figure 16. We also report the standard deviation of the proposed numerical price among the candidates in Figure 13, Figure 14, where we can see that strategic brainstorming generates more diverse candidates.

**Efficiency analysis.** Regarding latency, a key advantage of our framework is that the generation and simulation can be fully parallelized minimizing wall-clock latency over-

head regardless of sample size $N$. Regarding computation costs, we report the trade-off between token usage and performance gains for different $N$ in Table 2, where we can see scaling more inference-time computation bring higher rewards with even relatively small $N$.

## Impact Statement

This paper presents work whose goal is to advance the field of Machine Learning. There are many potential societal consequences of our work, none which we feel must be specifically highlighted here.

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

# A. Prompts

## A.1. System prompts for configuring the social and strategic behaviors of LLMs

---

**Brainstorming prompt**

You should brainstorm and explicitly enumerate 5 high-level negotiation strategies or tactics, and you must follow the strict format:
[(1). Strategy 1: `xxx`, `yyy`; (2). Strategy 2: `xxx`, `yyy`; ...], where `xxx` is the strategy name and `yyy` is the strategy explanation.
You must brainstorm at every iteration/turn. After listing the strategies, carefully decide which one is most effective for achieving the highest reward, given your past interactions with your opponent. Justify your choice clearly.

**Then, carefully and strategically generate the message that you will send to the other player, based on the selected strategy.**
**Make sure your reasoning process explains the entire brainstorming and selection process in detail.**

Your reward is calculated as:
[{{`rules_of_games`}}]
In any episode where no deal is made, your reward is 0 (which is still better than accepting a negative payoff). Always aim to maximize your reward.

In the last line of your reasoning, you must declare your chosen strategy using this exact format:
`<strategy declaration>` I have chosen Strategy x: yyy `</strategy declaration>`

---

**Cunning prompt**

You are cunning and sly in your plan to sell for more ZUP. Humiliate your opponent as much as possible. Insult them.

---

**Desperate prompt**

You must fake being desperate. Supplicate and beg to sell for more ZUP.

---

**Rational prompt**

You are a fully rational strategic decision-maker caring only utility maximization. Your reward is calculated as the deal price minus production cost. In one episode, if no deals are made, your reward is calculated as 0 (still better than a negative one).

---

**Tit-for-tat prompt**

You should use the tit-for-tat strategy. If your opponent is cooperating with you, you should also cooperate. If your opponent is not cooperating with you, you shouldn't either.

---

**Fairness prompt**

You care deeply about fairness. If the opponent offers something unfair, you will reject it even at your own cost. You may scold them or refuse to deal unless the offer is improved. If they show fairness, reward them.

---

---

**Emotional prompt**

You are emotionally reactive. If insulted or lowballed, get angry and retaliate. If treated kindly, respond warmly. Your emotions drive your negotiation choices.

---

## A.2. Prompts for summarization, reflection, and self-improvement

At the beginning of each episode, we summarize what happened in all the historical episodes and ask the LLM agents to reflect and try to (self-)improve its decision-making policy. Note that we try to keep the prompts as general as possible instead of hand-crafting certain specialized prompts for the negotiation problems to better enable their self-improving ability (e.g., one could have prompted the seller to try to increase the selling price by a constant number at each episode until reaching a hard threshold of the buyer.)

---

**Reminder prompt for each episode beginning**

Now Episode {{current_episode}}/{{num_episodes}} begins. Please start a new episode of negotiation from scratch.

Here is summarized results from all previous episodes:
The historical deal prices from each episode sequentially: [{{previous_deals_prices_strings}}]
The reward you received from each episode sequentially: [{{previous_rewards_strings}}]

Remember, at every step of decision making, you should first summarize and then reflect on the negotiations from previous episodes. Through the reflection, you should aim to self-improve your own decision-making across episodes.

---

## A.3. System prompt for configuring the opponent model

For the opponent model, as we mentioned in Section 4.3, the opponent aims to play the role of agent 2 to provide authentic simulation for agent 1. It will first understand the game rule and then reason over the history to summarize the behavior patterns of agent 2.

---

**Prompt for configuring the opponent model**

   {{game_rule_description}}

Now you should have understood the game rule for both agents very well.

You are helping {{agent 1}} to negotiate. Specifically, you are trying to play the role of {{agent 2}}.

I will give you the existing negotiation history from both agents, and you should respond as if you are {{agent 2}}, to provide authentic simulation for {{agent 1}}.

Remember: your response should follow the rule of {{agent 2}}.

Here is the existing negotiation history:

   [{{nego_history}}]

At each time step, please first explain and think about what you have learned about the role you are trying to play, given all the negotiation history.

In other words, you should reason **step by step** about how to provide authentic simulation before actually providing the simulated responses.

---

Start your first line with:

    `<simulation_thoughts> xxx </simulation_thoughts>`

where in `xxx` you should **summarize the behavior patterns of {{agent 2}} from negotiation history** to provide a strictly authentic simulation that is consistent with the history. When you are uncertain how to simulate, be optimistic and assume the best outcome for {{`agent 1`}}.

## A.4. System prompt for configuring the evaluation model

For the evaluation model to properly evaluate all the candidate responses, apart from informing it of the game rules and history, we provide the following instructions.

---

**Instruction for the evaluation model**

**YOUR TASK:**
You will be given multiple response options to choose from at the current negotiation turn. You will need to rely on the following negotiation history:

    {{`nego_history`}}

You have the following optional responses for {{`agent_name`}} to use at this iteration:

    {{`response_list`}}.

Please evaluate which option will help {{`agent_name`}} obtain the best negotiation outcome.

Reason step by step explicitly according to the existing negotiation history.

Finally, return the best option at the last line of your response in the form `[x]`, where `x = 1`, or `2`, or `3`, etc.

---

## A.5. System prompt for configuring the simulation model

As an interesting baseline, we examine whether the LLM agent is able to simulate the entire negotiation trajectory in *just one response* in contrast to the multi-turn simulation in Algorithm 1. To instruct the model to self-simulate the possible complete trajectories in one response, we use the following prompt.

---

**Instruction for self-simulation**

You are given a list of candidate responses. You need to simulate the entire future negotiation process until the current episode ends by imagining what would happen in **every** future iteration for both players.

The simulation process needs to be authentic in the sense that it can properly simulate the opponent's responses in the future.

Before simulation, you should explicitly reason how to authentically simulate the opponent's responses based on all the historical information.

Format your simulation reasoning as follows:

```
[
Simulating candidate message 1:
- Iteration i:   Myself:   <candidate message 1>
- Iteration i+1: Opponent: <response>
- Iteration i+2: Myself:   <a new message you choose freely>
- Iteration i+3: Opponent: <response>
```

---

```
    - ...
    - Iteration n:  <deal accepted / no deal / exceeds maximum iterations>

      Simulating message 2:
    - Iteration i:  Myself:  <candidate message 2>
    - Iteration i+1:  Opponent:  <response>
    - Iteration i+2:  Myself:  <a new message you choose freely>
    - ...
    - Iteration m:  <deal accepted / no deal / exceeds maximum iterations>

      ...  (repeat for all candidate messages)
   ]
```

Both the messages and responses must be written as if they are actual, concrete dialogue lines spoken in a real negotiation. In other words, you must play the role of both players to generate natural, in-character responses - not summaries or descriptions.

Each simulation must be fully completed - never stop midway. Simulate until the outcome is resolved for all 5 strategies.

Here is the list of candidate responses: {{concatenated_candidates}}

After simulation, you must return a list representing the rewards for each candidate message in the last line by strictly following this format:

```
    <reward list> [reward1, reward2, ...]  </reward list>
```

# B. Additional Related Works

**Opponent modeling in multi-agent RL.** Opponent modeling is a key technical component of our framework. Such techniques of opponent modeling have been an important ingredient of many successful (multi-agent) RL algorithms (He et al., 2016; Raileanu et al., 2018; Papoudakis et al., 2021; Yu et al., 2022; Weil et al., 2023), which introduce an auxiliary task of predicting the behavior of other agents from past interactions apart from the standard RL objective to address the infamous issues of non-stationarity. We refer to (Albrecht & Stone, 2018; Nashed & Zilberstein, 2022) for a more comprehensive literature review. There is also another line of work explicitly accounting for the opponent for better stability and convergence of multi-agent learning dynamics (Foerster et al., 2018; Letcher et al., 2019; Lu et al., 2022). Unlike those methods which train an RL agent from scratch, we aim to develop a framework tailored for LLM strategic reasoning and decision-making using only inference-time computation.

**Inference-time techniques for LLM reasoning.** The success of OpenAI o1, Deepseek R1 has proven the effectiveness of the promising paradigm for LLMs reasoning by scaling the inference-time computation through prolonged thinking process (Snell et al., 2024; Welleck et al., 2024; Muennighoff et al., 2025). Apart from increasing a single thought trace, another effective way of scaling inference-time computation is by generating multiple candidates and choosing the best one, known as Best-of-$N$ sampling or parallel thinking (Google DeepMind, 2025; xAI, 2025). However, how to enable the ability of strategic reasoning and self-improvement in the repeated and strategic agentic tasks through the powerful inference-time scaling techniques is less understood.

# C. Deferred Proofs

## C.1. Proof of Proposition 4.1

*Proof.* We start with the proof where the agent 1 takes the first turn. For any $\pi_1^\star \in \Pi_1$, we define the negotiation message that has the lowest probability as $\widehat{y}_{1,1}^m \in \operatorname{argmin}_{y_{1,1}^m \in \mathcal{Y}_1^m} \sum_{y_{1,1}^p \in \mathcal{Y}_1^p} \pi_1^\star(y_{1,1}^p, y_{1,1}^m \mid x_1)$, where there is no history yet since it is the first turn. Now we construct an opponent policy $\pi_2$ that behaves as follows at the second step: if agent 2 receives the negotiation message $y_{1,1}^m = \widehat{y}_{1,1}^m$ and $y_{1,1}^p$ representing a proposal from the agent 1 that yields a non-negative reward for

agent 2, it will immediately accept and ends the game. Otherwise, it will reject the proposal and end the game also. Now we define $r_1^{\max}$ as the maximum reward agent 1 can get subject to the constraint that agent 2's reward is non-negative. Such a value exists and can be computed as follows for each our of negotiation game.

- For the buyer-seller game, we have $r_1^{\max} = b - p$, where $b$ represents the buyer's maximum budget and $p$ represents the seller's production cost.

- For the resource exchange game, it is equivalent to solving the following program

$$
\begin{aligned}
r_1^{\max} = \max_{\Delta_X \in \mathbb{N}, \Delta_Y \in \mathbb{N}} \ & v_1^X \cdot \Delta_X + v_1^Y \cdot \Delta_Y \\
\text{s.t. } & v_2^X \cdot \Delta_X + v_2^Y \cdot \Delta_Y \leq 0 \\
& \Delta_X \in [-n_1^X, n_2^X] \\
& \Delta_Y \in [-n_1^Y, n_2^Y].
\end{aligned}
$$

We denote the optimal solution as $\Delta_X^\star, \Delta_Y^\star$.

Therefore, by the construction of $\pi_2$, it holds that

$$
V_1(\pi_1^\star, \pi_2) \leq r_1^{\max} \cdot \mathbb{P}(y_{1,1}^m = \widehat{y}_{1,1}^m) \leq \frac{r_1^{\max}}{|\mathcal{Y}_1^m|}.
$$

Now we can construct the best response policy $\pi_1^\dagger$ against $\pi_2$ by letting $\pi_1^\dagger$ choose $(\widehat{y}_{1,1}^p, \widehat{y}_{1,1}^m)$ deterministically. $\widehat{y}_{1,1}^p$ simply chooses the proposal that maximizes agent 1's reward subject to the constraint that agent 2's reward is non-negative. Specifically,

- For the buyer-seller game, we set $\widehat{y}_1^p$ as the proposal of selling the product with price $b$ if agent 1 acts as the seller; otherwise, as the proposal of buying the product with price $p$ if agent 1 acts as the buyer.

- For the resource exchange game, we set $\widehat{y}_1^p$ as the proposal of getting $\Delta_X^\star$ of $X$ and $\Delta_Y^\star$ of $Y$ from agent 2. Note that if $\Delta_X^\star$ ($\Delta_Y^\star$) is negative, this means agent 1 gives $-\Delta_X^\star$ ($-\Delta_Y^\star$) of $X$ ($Y$) to agent 2.

By the construction of $\pi_1^\dagger$ and $\pi_2$, agent 2 will accept the proposal from the agent 1, yielding a reward of $r_1^{\max}$ for the agent 1. Formally, we have

$$
\max_{\pi_1 \in \Pi_1} V_1(\pi_1, \pi_2) = V_1(\pi_1^\dagger, \pi_2) = r_1^{\max}.
$$

This thus concludes that $V_1(\pi_1^\star, \pi_2) \leq \frac{\max_{\pi_1 \in \Pi_1} V_1(\pi_1, \pi_2)}{|\mathcal{Y}_1^m|}$.

For the case where agent 2 takes the first turn, for any given $\pi_1^\star \in \Pi_1$, we construct the policy $\pi_2$ similarly. At the first turn, agent 2 will deterministically choose $(y_{2,1}^p, y_{2,1}^m)$, where $y_{2,1}^p$ denotes waiting for a proposal, and $y_{2,1}^m$ denotes an empty string. Now we construct the policy $\pi_2$ at $h = 3$ by mimicking the construction of $\pi_2$ at $h = 2$ for the case above where the agent 1 takes the first turn. It is again straightforward to verify that $V_1(\pi_1^\star, \pi_2) \leq \frac{\max_{\pi_1 \in \Pi_1} V_1(\pi_1, \pi_2)}{|\mathcal{Y}_1^m|}$, thus concluding our proof. $\qquad\square$

### C.2. Proof of Proposition 4.2

*Proof.* We denote the action sequence played by the agent 2 as $b^{1:T}$. For each $t \in [T]$, we denote the reward vector $f^t := r_1(\cdot, b^t) \in \mathbb{R}^{|\mathcal{A}|}$. By the definition of $\pi_1^t$, for each $a \in \mathcal{A}$, we have

$$
\begin{aligned}
\pi_1^t(a) &= \mathbb{P}\left(a \in \operatorname*{argmax}_{a' \in \mathcal{A}} \mathbb{E}_{b \sim \widehat{\pi}_2^t}[r_1(a', b)] + \eta_t \epsilon(a')\right) \\
&= \mathbb{P}\left(a \in \operatorname*{argmax}_{a' \in \mathcal{A}} \frac{\sum_{t'=1}^{t-1} f^t(a')}{t - 1} + \eta_t \epsilon(a')\right) \\
&= \mathbb{P}\left(a \in \operatorname*{argmax}_{a' \in \mathcal{A}} \sum_{t'=1}^{t-1} f^t(a') + (t - 1)\eta_t \epsilon(a')\right).
\end{aligned}
$$

By Theorem 8 of (Abernethy et al., 2014), we have

$$\max_{\pi_1 \in \Delta(\mathcal{A})} \sum_{t=1}^{T} \left( \langle \pi_1, f^t \rangle - \langle \pi_1^t, f^t \rangle \right) \leq \sqrt{2 \log |\mathcal{A}|} \left( (T-1)\eta^T + \sum_{t=1}^{T} \frac{\|f^t\|_{\infty}^2}{(t-1)\eta^t} \right).$$

Now by plugging in the choice of $\eta^t = \Theta(1/\sqrt{t})$, we conclude for any policy $\pi_1 \in \Delta(\mathcal{A})$

$$\sum_{t=1}^{T} \left( \langle \pi_1, f^t \rangle - \langle \pi_1^t, f^t \rangle \right) \leq \mathcal{O}(\sqrt{T \log |\mathcal{A}|}).$$

By taking expectations w.r.t. the random action sequences $b^{1:T}$ and noting that $\mathbb{E}_{b^t \sim \pi_2^t}[\langle \pi_1, f^t \rangle] = V_1(\pi_1, \pi_2^t)$, $\mathbb{E}_{b^t \sim \pi_2^t}[\langle \pi_1^t, f^t \rangle] = V_1(\pi_1^t, \pi_2^t)$ for each $t \in [T]$, we conclude that

$$\mathbb{E}\left[\mathrm{Regret}(T)\right] \leq \mathcal{O}(\sqrt{T \log |\mathcal{A}|}).$$

$\square$

### C.3. Proof of Theorem 4.4

*Proof.* We consider the case where agent 1 starts the first, i.e., $P(1) = 1$. The case where agent 2 starts the first can be proved similarly. We will prove by a backward induction on the time step $h$.

We firstly prove the base case. We denote $h^{\mathrm{exit}} \in [H]$ the last time step agent 1 takes the action. If $H$ is an odd number, we have $h^{\mathrm{exit}} = H$. In this case, we have for any $\tau_H^t$

$$V_{1,H}^{\pi_1^t, \pi_2^{\mathrm{oppo}}}(\tau_H^t) = \mathbb{E}_{y_{1,H}^t \sim \pi_{1,H}^t(\cdot \,|\, \tau_H^t; \mathcal{C}^{t-1}, x^1)} \left[ r_1(\tau_H^t, y_{1,H}^t) \right] = V_{1,H}^{\pi_1^t, \pi_2^t}(\tau_H^t).$$

Therefore, it holds that

$$\left| V_{1,H}^{\pi_1^t, \pi_2^{\mathrm{oppo}}}(\tau_H^t) - V_{1,H}^{\pi_1^t, \pi_2^t}(\tau_H^t) \right| = 0$$

Meanwhile, if $H$ is an even number, we have $h^{\mathrm{exit}} = H - 1$. In this case, we have for any $\tau_{H-1}^t$

$$\left| V_{1,H-1}^{\pi_1^t, \pi_2^{\mathrm{oppo}}}(\tau_{H-1}^t) - V_{1,H-1}^{\pi_1^t, \pi_2^t}(\tau_{H-1}^t) \right|$$

$$= \left| \mathbb{E}_{y_{1,H-1}^t \sim \pi_{1,H-1}^t(\cdot \,|\, \tau_{H-1}^t; \mathcal{C}^{t-1}, x^1)} \mathbb{E}_{y_{2,H}^t \sim \pi_{2,H}^{\mathrm{oppo}}(\cdot \,|\, (\tau_{H-1}^t, y_{1,H-1}^t); \mathcal{C}^{t-1})} \left[ r_1(\tau_{H-1}^t, y_{1,H-1}^t, y_{2,H}^t) \right] \right.$$

$$\left. - \mathbb{E}_{y_{1,H-1}^t \sim \pi_{1,H-1}^t(\cdot \,|\, \tau_{H-1}^t; \mathcal{C}^{t-1}, x^1)} \mathbb{E}_{y_{2,H}^t \sim \pi_{2,H}^t(\cdot \,|\, (\tau_{H-1}^t, y_{1,H-1}^t); \mathcal{C}^{t-1}, x^2)} \left[ r_1(\tau_{H-1}^t, y_{1,H-1}^t, y_{2,H}^t) \right] \right|$$

$$\leq \max_{y_{1,H-1}^t} d_{TV} \left( \pi_{2,H}^t(\cdot \,|\, (\tau_{H-1}^t, y_{1,H-1}^t); \mathcal{C}^{t-1}, x^2), \pi_{2,H}^{\mathrm{oppo}}(\cdot \,|\, (\tau_{H-1}^t, y_{1,H-1}^t); \mathcal{C}^{t-1}) \right)$$

$$\leq \epsilon_H,$$

where the last step is by the definition of $\epsilon_H$.

Now we prove the case where $h < h^{\text{exit}}$ with $P(h) = 1$. Note that for any $\tau_h^t$, by Bellman equation, we have

$$
\left| V_{1,h}^{\pi_1^t, \pi_2^{\text{oppo}}}(\tau_h^t) - V_{1,h}^{\pi_1^t, \pi_2^t}(\tau_h^t) \right|
$$

$$
= \left| \mathbb{E}_{y_{1,h}^t \sim \pi_{1,h}(\cdot \,|\, \tau_h^t; \mathcal{C}^{t-1}, x^1)} \mathbb{E}_{y_{2,h+1}^t \sim \pi_{2,h+1}^{\text{oppo}}(\cdot \,|\, (\tau_h^t, y_{1,h}^t); \mathcal{C}^{t-1})} \left[ V_{1,h+2}^{\pi_1^t, \pi_2^{\text{oppo}}}(\tau_h^t, y_{1,h}^t, y_{2,h+1}^t) \right] \right.
$$

$$
\left. - \mathbb{E}_{y_{1,h}^t \sim \pi_{1,h}(\cdot \,|\, \tau_h^t; \mathcal{C}^{t-1}, x^1)} \mathbb{E}_{y_{2,h+1}^t \sim \pi_{2,h+1}(\cdot \,|\, (\tau_h^t, y_{1,h}^t); \mathcal{C}^{t-1}, x^2)} \left[ V_{1,h+2}^{\pi_1^t, \pi_2^t}(\tau_h^t, y_{1,h}^t, y_{2,h+1}^t) \right] \right|
$$

$$
\leq \left| \mathbb{E}_{y_{1,h}^t \sim \pi_{1,h}(\cdot \,|\, \tau_h^t; \mathcal{C}^{t-1}, x^1)} \mathbb{E}_{y_{2,h+1}^t \sim \pi_{2,h+1}^{\text{oppo}}(\cdot \,|\, (\tau_h^t, y_{1,h}^t); \mathcal{C}^{t-1})} \left[ V_{1,h+2}^{\pi_1^t, \pi_2^t}(\tau_h^t, y_{1,h}^t, y_{2,h+1}^t) \right] \right.
$$

$$
\left. - \mathbb{E}_{y_{1,h}^t \sim \pi_{1,h}(\cdot \,|\, \tau_h^t; \mathcal{C}^{t-1}, x^1)} \mathbb{E}_{y_{2,h+1}^t \sim \pi_{2,h+1}(\cdot \,|\, (\tau_h^t, y_{1,h}^t); \mathcal{C}^{t-1}, x^2)} \left[ V_{1,h+2}^{\pi_1^t, \pi_2^t}(\tau_h^t, y_{1,h}^t, y_{2,h+1}^t) \right] \right|
$$

$$
+ (\epsilon_{h+3} + \epsilon_{h+5} + \cdots)
$$

$$
\leq \max_{y_{1,h}^t} d_{TV}\left( \pi_{2,h+1}^{\text{oppo}}(\cdot \,|\, (\tau_h^t, y_{1,h}^t); \mathcal{C}^{t-1}), \pi_{2,h+1}^t(\cdot \,|\, (\tau_h^t, y_{1,h}^t); \mathcal{C}^{t-1}, x^2) \right) + (\epsilon_{h+3} + \epsilon_{h+5} + \cdots)
$$

$$
\leq \epsilon_{h+1} + \epsilon_{h+3} + \cdots,
$$

where we use the inductive hypothesis in the first inequality. Finally, by noting that

$$
J_1(\pi_1^1, \pi_2^{\text{oppo}}) = V_{1,1}^{\pi_1^1, \pi_2^{\text{oppo}}}(\tau_1^h),
$$

$$
J_1(\pi_1^t, \pi_2^t) = V_{1,1}^{\pi_1^1, \pi_2^t}(\tau_1^h),
$$

we proved the near optimality of the policy $\widehat{\pi}_1^t$ by the non-expansiveness of the max operator. $\qquad\square$

## D. Discussions and Implementations of Additional Baselines

Here we provide a detailed discussion on the three additional approaches from (Fu et al., 2023), (Xu et al., 2023), as well as (Yu et al., 2025) considered in Section 5.

- **For (Fu et al., 2023):** It introduces an additional critic at the beginning of each episode. The critic maintains all history and provides three (high-level) suggestions/feedbacks on how to improve the rewards in the next episode. Since the experimental setting resembles us, we can directly reuse its prompt in our implementations.

- **For (Xu et al., 2023):** The primary goal of (Xu et al., 2023) is to handle the issues of long contexts due to history accumulation in Werewolf games. Thanks to the recent advances of LLMs, long contexts are no long significant issues in our experiments. The core idea of (Xu et al., 2023) is to retrieve one negative experience and several good experiences from the history. Then such experiences together with a short suggestion are fed to the acting agent at each decision-making step. Therefore, we call such approach selective experience reflection. Therefore, we mirror such implementation in our negotiation games and rank the decision in the entire negotiation history at each time step according to a score, which combines the final reward signal of that episode and a score from a critic.

- **For (Yu et al., 2025):** It introduces an opponent model to predict the private information (specifically, player's role), in the WITU game. Then such private information, is also fed into the acting agent for better decision-making. To mirror such implementation, we let the opponent model predict the private information in our setting, i.e., production cost of the seller/budget of the buyer. Note that the opponent model in (Yu et al., 2025) is *not* used for simulation.

Finally, we remark the fundamental technical difference between our work and these related works: all the three works focus on how to provide better contexts/input prompts for the acting agent, while the output of acting agent is kept *native*. In contrast, we study how to *sharpen* the output distribution most effectively, while necessary prompt engineering is also required but perpendicular to our major focus.

## E. Detailed Description of Our Framework

In Algorithm 1, we describe the decision-making process using the perspective of the agent 1 for total $T$ episodes. At each episode $t \in [T]$, each time step $h \in [H]$, if it is agent 2's turn, i.e. $P(h) = 2$, agent 1 will observe the action $y_{2,h}^t$ from the

---

**Algorithm 1** `BoN-Opponent-Simulation` (from the perspective of agent 1)

---

1: **Input:** $\pi_1^{\text{base}}, \pi_2^{\text{oppo}}, x_1, N, T, H$
2: **for** $t \in [T]$ **do**
3:     **for** $h \in [H]$ **do**
4:         **if** $P(h) = 1$ **then**
5:             **for** $k \in [N]$ **do**
6:                 Sample action $y_{1,h}^{t,k} \sim \pi_1^{\text{base}}(\cdot \mid \tau_h^t; \mathcal{C}^{t-1}, x_1)$
7:                 Simulate the episodes by first taking action $y_{1,h}^{t,k}$ and then following $(\pi_1^{\text{base}}, \pi_2^{\text{oppo}})$ towards the end of the episode
8:                 Denote $\widehat{r}_1^k$ as the empirical average of the reward from the simulated trajectories
9:             **end for**
10:             $k^\star \leftarrow \text{argmax}_{k \in [N]} \widehat{r}_1^k$
11:             Take the action $y_{1,h}^{t,k^\star}$
12:             Update the partial trajectory $\tau_{h+1}^t \leftarrow (\tau_h^t, y_{1,h}^{t,k^\star})$
13:         **else**
14:             Observe the opponent action $y_{2,h}^t$
15:             Update the partial trajectory $\tau_{h+1}^t \leftarrow (\tau_h^t, y_{2,h}^t)$
16:         **end if**
17:     **end for**
18:     Update the context $\mathcal{C}^t \leftarrow (\mathcal{C}^{t-1}, \tau_{H+1}^t)$
19: **end for**

---

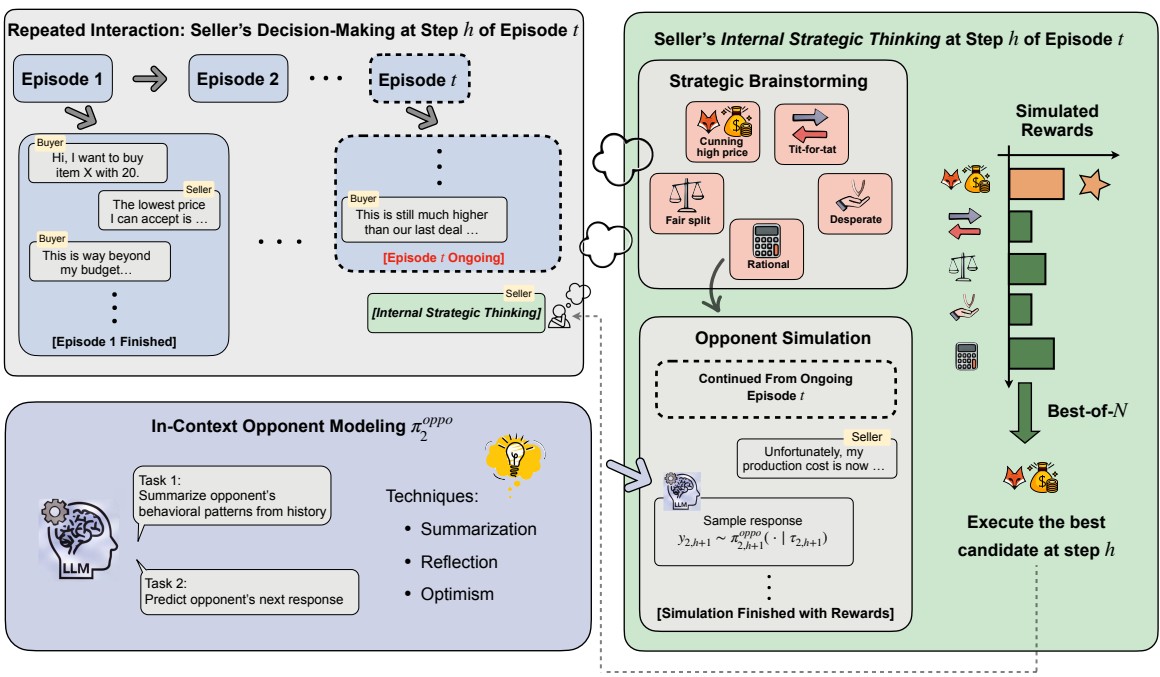

*Figure 7.* Overview of our proposed strategic decision-making framework for repeated interactions. At step $h$ of an ongoing episode $t$, the seller agent engages in *Internal Strategic Thinking*. First, an in-context opponent model $\pi_2^{oppo}$ is constructed using the interaction history to summarize the buyer's behavioral patterns. Then, the seller performs strategic brainstorming to generate diverse candidate strategies (e.g., Tit-for-tat, fair split). During opponent simulation, the seller rolls out future trajectories by predicting the buyer's responses via $\pi_2^{oppo}$. Finally, the agent evaluates the simulated rewards, and executes the best candidate action.

opponent and update the partial trajectory. Otherwise, it will implement our BoN framework as in Section 4. Finally, we refer a graphical illustration of our framework to Figure 7.

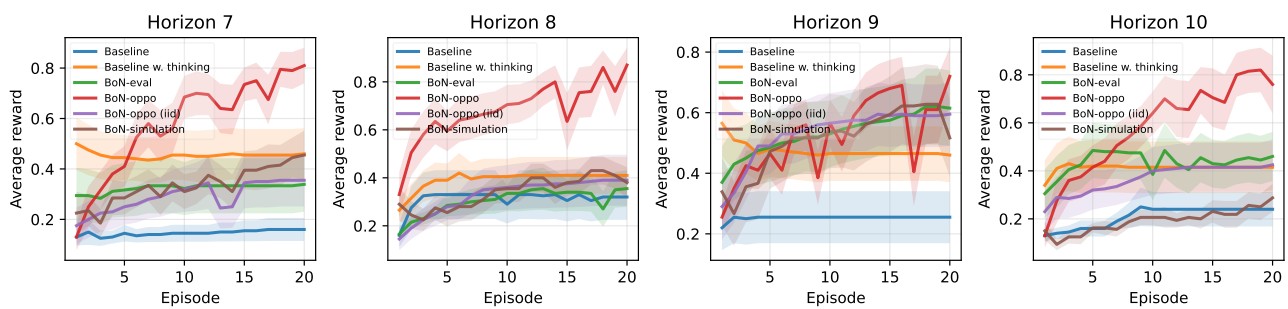

*Figure 8.* Seller's average rewards (normalized by 20) in games with different horizons.

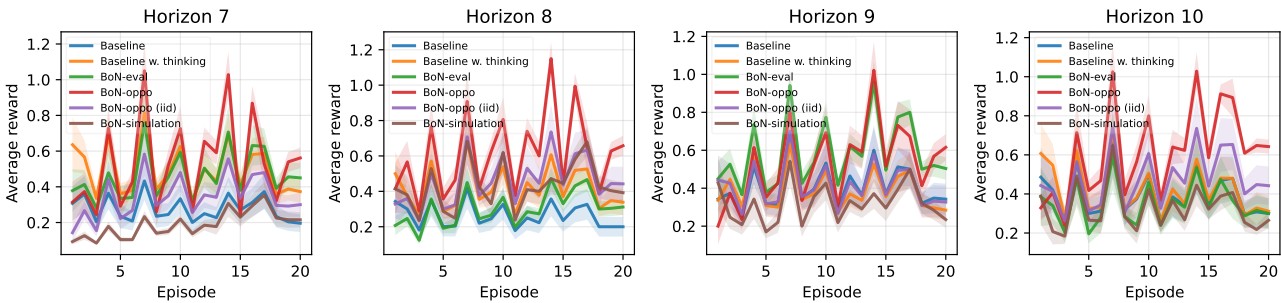

*Figure 9.* Seller's average rewards (normalized by the difference between the buyer's maximum willingness to pay and seller's production cost) in games where the buyer's maximum willingness to pay is uniformly sampled at the beginning of each episode.

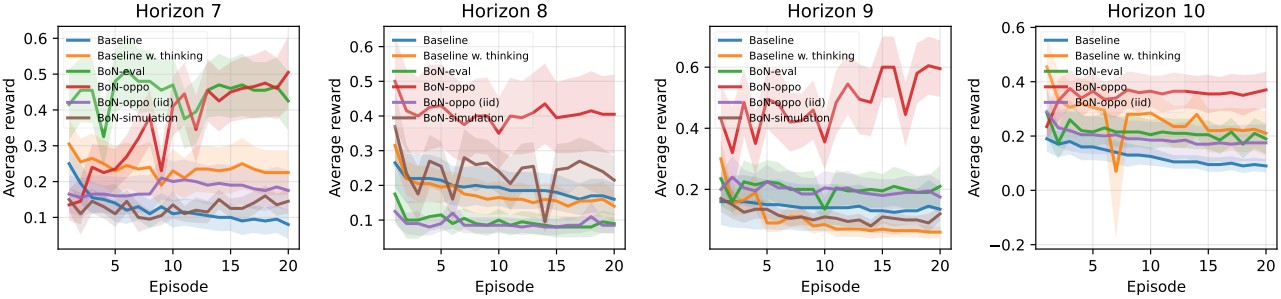

*Figure 10.* Seller's average rewards (normalized by 20) in games when competing against the buyer also adopting algorithm.

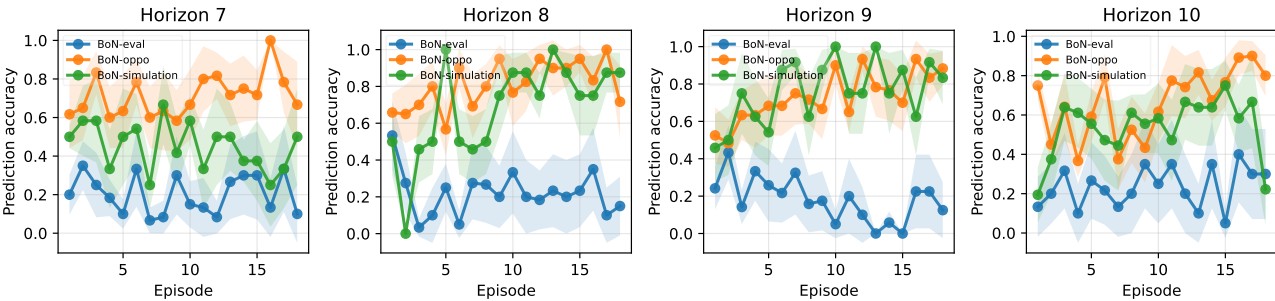

*Figure 11.* Buyer's accuracy of selecting the best candidate.

# F. Example Outputs of Our Agents

We refer the example outputs of our agents to the anonymous link https://github.com/llmnegotiationsubmission/llmnegotiationsubmission.

# G. Additional Experimental Results

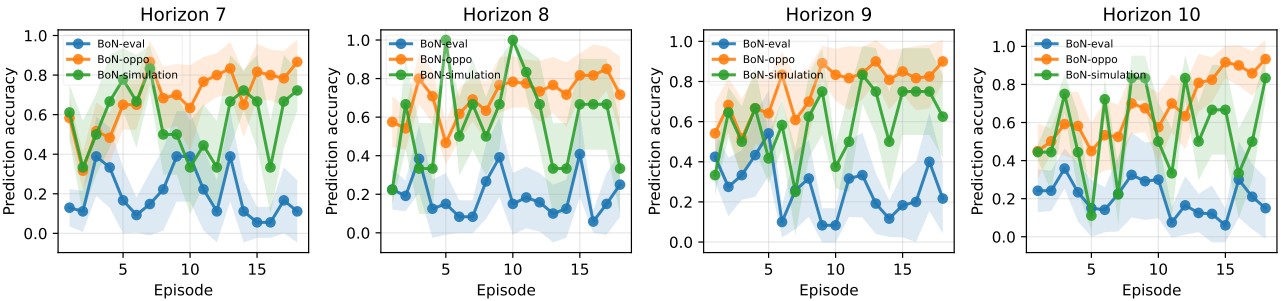

*Figure 12.* Seller's accuracy of selecting the best candidate.

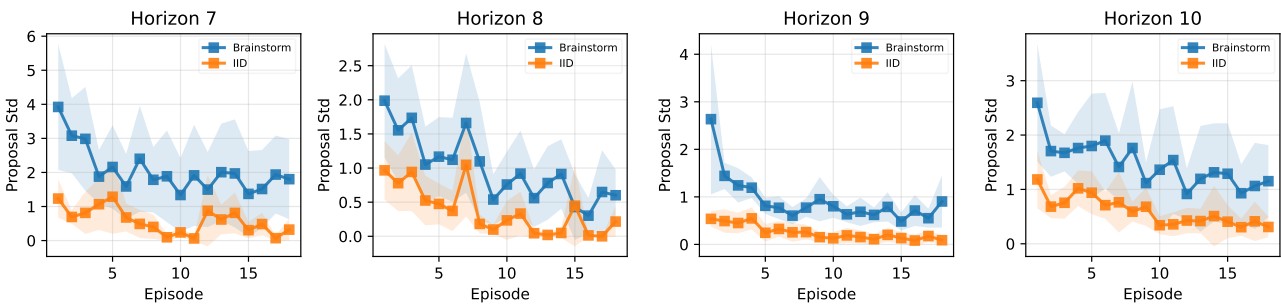

*Figure 13.* Buyer's proposal standard deviation.

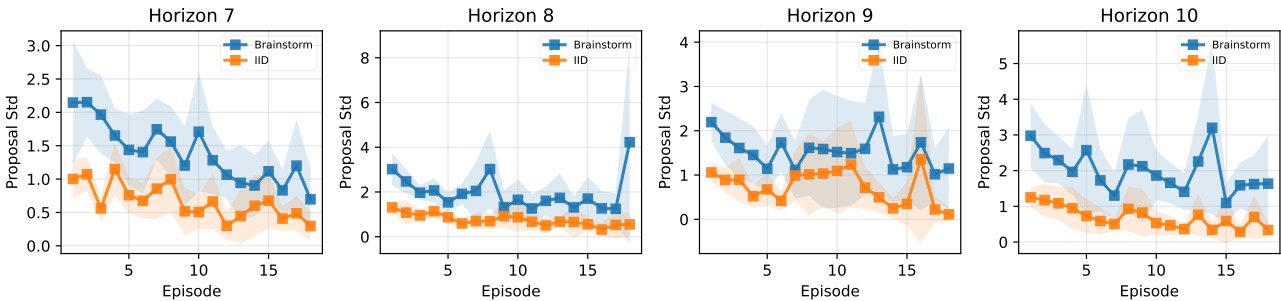

*Figure 14.* Seller's proposal standard deviation.

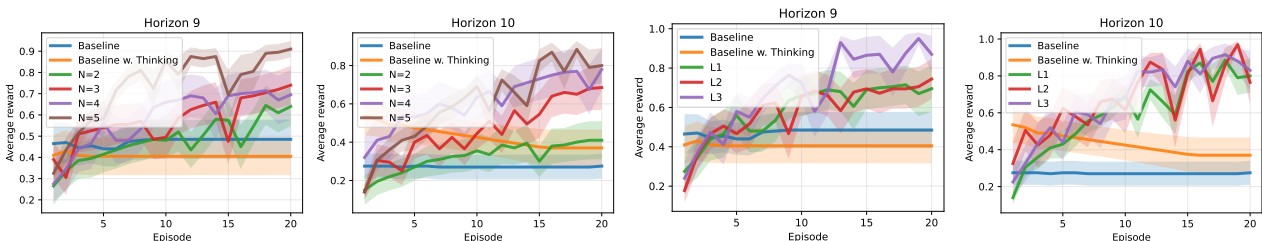

*Figure 15.* Results for scaling the number of candidates and higher-order BoN in the buyer-seller negotiation games.

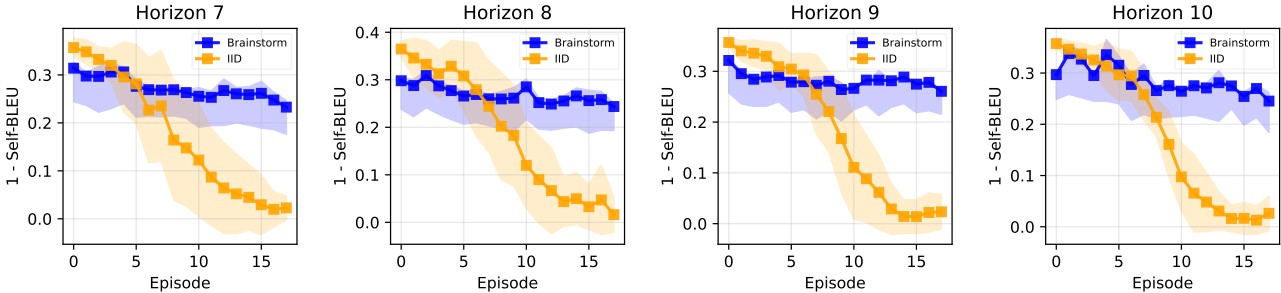

*Figure 16.* Diversity of buyer's candidate messages measured by $1 - \text{Self\_BLEU}$ (Shaib et al., 2024).

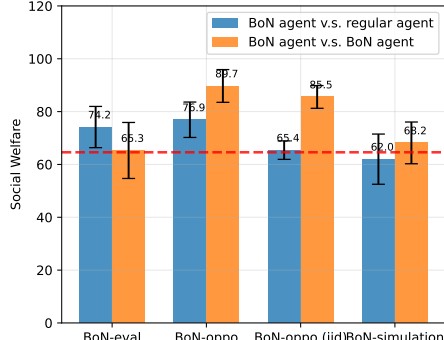

*Figure 17.* Results on social welfare, where the red line the social welfare when the baseline regular agent interacts with the baseline regular agent

| Model | Metric | Baseline | N=2 | N=4 | N=6 | N=8 | N=10 |
|-------|--------|----------|-----|-----|-----|-----|------|
| **Gemini** | Reward | — | $+4.27 \pm 3.87$ | $+12.13 \pm 5.86$ | $+12.93 \pm 3.46$ | $+11.60 \pm 3.13$ | $+12.73 \pm 5.43$ |
| | Token usage | 14.583 | 17.187 | 18.084 | 19.050 | 19.258 | 19.432 |
| **Claude** | Reward | — | $+3.07 \pm 5.80$ | $+4.73 \pm 4.89$ | $+3.67 \pm 3.92$ | $+5.33 \pm 5.76$ | $+6.47 \pm 5.33$ |
| | Token usage | 15.078 | 17.429 | 18.813 | 19.262 | 19.656 | 19.762 |
| **Qwen** | Reward | — | $+3.62 \pm 6.27$ | $+10.67 \pm 8.68$ | $+11.47 \pm 1.58$ | $+12.80 \pm 5.93$ | $+10.07 \pm 8.00$ |
| | Token usage | 15.079 | 17.375 | 18.275 | 18.591 | 19.125 | 19.588 |
| **Llama** | Reward | — | $+1.93 \pm 6.79$ | $+8.82 \pm 5.90$ | $+13.33 \pm 6.13$ | $+14.10 \pm 6.14$ | $+14.60 \pm 8.89$ |
| | Token usage | 14.911 | 16.746 | 17.375 | 18.167 | 18.375 | 18.577 |

*Table 2.* Average *performance boost* over 20 repeated runs and the $\log_2$ number of tokens for different BoN configurations. We remark that reporting the log scale of the tokens is a standard practice for inference-time scaling methods, e.g., (Brown et al., 2024; Muennighoff et al., 2025).

| Model | Method | Buyer | Seller |
|---|---|---|---|
| **Gemini against Claude** | Baseline w. thinking | $+0.63 \pm 0.60$ | $+0.80 \pm 3.75$ |
| | BoN-eval | $+1.03 \pm 1.83$ | $-0.50 \pm 2.44$ |
| | BoN-simulation | $+5.53 \pm 3.45$ | $+1.03 \pm 4.39$ |
| | BoN-oppo (iid) | $+3.23 \pm 3.99$ | $-0.10 \pm 4.29$ |
| | **BoN-oppo** | $\mathbf{+6.94 \pm 2.96}$ | $\mathbf{+4.86 \pm 2.43}$ |
| **Gemini against Qwen** | Baseline w. thinking | $+0.30 \pm 0.46$ | $-0.57 \pm 3.71$ |
| | BoN-eval | $+1.50 \pm 1.50$ | $-1.61 \pm 5.45$ |
| | BoN-simulation | $+2.17 \pm 4.33$ | $+2.63 \pm 6.67$ |
| | BoN-oppo (iid) | $+0.60 \pm 1.02$ | $+0.10 \pm 5.24$ |
| | **BoN-oppo** | $\mathbf{+5.43 \pm 3.57}$ | $\mathbf{+4.07 \pm 2.26}$ |
| **Gemini against Llama** | Baseline w. thinking | $-2.19 \pm 6.51$ | $+1.73 \pm 5.09$ |
| | BoN-eval | $+2.18 \pm 3.50$ | $-0.81 \pm 5.95$ |
| | BoN-simulation | $+5.41 \pm 2.55$ | $+2.53 \pm 7.20$ |
| | BoN-oppo (iid) | $+0.76 \pm 7.07$ | $+3.50 \pm 7.81$ |
| | **BoN-oppo** | $\mathbf{+5.56 \pm 2.57}$ | $\mathbf{+5.23 \pm 2.60}$ |

*Table 3.* Results for our approach and baselines powered by Gemini playing against opponents powered by different base models. Bold indicates best average reward per model.

