# OpenReview forum: "Scaling Inference-Time Computation via Opponent Simulation: Enabling Online Strategic Adaptation in Repeated Negotiation"
_ICML.cc/2026/Conference — ICML 2026 regular_

### Official Review · Reviewer_z1u3 · 2026-03-11

**Soundness:** 3
**Presentation:** 3
**Significance:** 3
**Originality:** 3
**Overall Recommendation:** 5
**Confidence:** 3

**Summary:**

This work studies how to enable online strategic adaptation for large language models in repeated negotiation games without parameter updates. The authors argue that static policies trained offline are often overly conservative and fail to exploit opponent-specific weaknesses in dynamic interactions. Inspired by smooth Fictitious Play, the authors propose to scale inference-time computation through a two-step pipeline: (1) in-context opponent modeling, where an auxiliary LLM summarizes and imitates the opponent’s time-averaged behavior, acting as a opponent simulator, and (2) best-of-N action selection via full trajectory simulation of the ego agent against the opponent model. By ranking candidate strategies based on the results of these simulated rollouts, the method effectively adds additional test-time computation into online strategic adaptation. Experiments on repeated buyer–seller and resource exchange games show consistent reward gains over baseline methods, including standard prompting, thinking-based baselines, and other inference-time scaling approaches. These results demonstrate that scaling compute in structured opponent simulation enables robust and principled online adaptation in multi-agent LLM settings.

**Compliance With Llm Reviewing Policy:**

Affirmed.

**Final Justification:**

Thank you for the additional clarification. This follow-up response more directly addresses my main remaining concern regarding the faithfulness of the prompt-based opponent model to the theoretical assumptions. Combined with the earlier rebuttal on compute scaling and broader-domain evaluation, I believe my main concerns have now been sufficiently addressed. I am therefore happy to raise my score from Weak Accept to Accept.

**Key Questions For Authors:**

1. How does the proposed role-playing opponent model perform compared with actually finetuned opponent model?
2. How sensitive is performance to opponent-model quality?
3. What is the compute–performance scaling behavior?
4. Does the method generalize beyond negotiation-style games?

**Limitations:**

1. Theoretical–practical gap. While the paper presents regret and error-propagation guarantees, these results rely on simplified assumptions (e.g., bounded total variation distance, well-defined action spaces) that may not strictly hold in LLM-based negotiation.
2. Heuristic opponent modeling. Belief formation is implemented via in-context summarization rather than explicit statistical modeling of opponent behavior.
3. Domain restriction to structured two-player games.

**Strengths And Weaknesses:**

This is a very solid piece of work that targets the online strategic adaptation of LLM agents in multi-agent settings. I would like to start by pointing out what I think is very well done.

Strengths:

1. The paper is very well motivated and gives a very clear problem framing. The authors clearly identify an important gap in current LLM-based decision-making systems, which is the lack of principled online adaptation in (repeated) strategic interactions. The authors point out the importance of such a gap by convincingly arguing that static policies (even equilibrium-based ones) can be overly conservative and suboptimal in repeated negotiation settings.
2. The paper is theoretically very well grounded. By introducing smooth Fictitious Play and min-regret learning, the paper provides a principled foundation for inference-time strategic adaptation. The theoretical analysis (including the regret guarantees and opponent-model error propagation bounds) strengthens the conceptual clarity of the approach and distinguishes it from purely heuristic/empirical prompt-based methods.
3. The algorithmic implementation is clean and interpretable. The proposed method decompos online adaptation into belief formation (in-context opponent modeling) and best response (BoN with opponent simulation) is intuitive and well-structured. This separation improves interpretability and makes the framework extensible to other strategic settings.
4. The method achieves online adaptation purely through test-time computation scaling, without gradient updates.
5. The paper demonstrates strong and consistent empirical results. Across repeated buyer–seller and resource exchange games, the method demonstrates consistent improvements over standard prompting, thinking-based baselines, BoN variants, and other adaptive methods. The evaluation includes robustness to stochastic environments and cross-model validation, which strengthens the empirical claims.

That being said, I do believe there are still a number of points that need further study or discussion.

Weaknesses:

1. While the paper presents heavy theoretical analysis, the actual implementation is relatively simple: in-context opponent summarization combined with BoN rollout simulation. The theoretical guarantees are largely introduced from classical sFP and do not tightly model the behavior of LLM agents. The gap between the theoretical model and the algorithm implementation remains noticeable.
2. This for me is the elephant in the room: belief formation is implemented via prompt-based summarization and role-playing rather than explicit empirical frequency tracking or calibrated density estimation. It is therefore unclear how faithfully the opponent model approximates the theoretical assumptions required by the regret analysis. It would be beneficial to see a empirical error analysis between the role-playing opponent model and actually finetuned opponent model.
3. Computational cost scaling is not deeply analyzed. While the paper demonstrates the effectiveness of dedicating additional compute in opponent modeling and BoN, the scaling effect is not studied for different compute budgets. A more systematic analysis of performance–compute trade-offs would strengthen the practical implications.
4. Limited domain diversity. Experiments are confined to two-player repeated negotiation environments. Although these are non-trivial language-based games, they remain structurally constrained. It is still unclear whether the framework generalizes to more complex multi-agent environments or open-ended agent tasks.

---

> ### Author Rebuttal · Authors · 2026-03-31
>
> We thank Reviewer z1u3 for the very positive feedback. We are greatly encouraged that the reviewer found this to be “a very solid piece of work.” We address the reviewer’s concerns as follows.
>
> ## Response to W1:
> We fully acknowledge that exactly implementing sFP is intractable due to the exponentially large action space, leaving a gap between the theoretical guarantees of the classical model and the empirical performance of practical LLM agents. This since theory is most useful for conveying fundamental ideas, rather than capturing every detail of a practical system. That said, we do not believe this gap undermines the value of our work, or more broadly, the line of research that uses classical principles to design well-grounded LLM methods. We highlight two concrete ways in which theory guides our empirical design.
>
> (1). sFP provides the theoretical basis for relying on opponent modeling, rather than sequential modeling, when the opponent evolves. In online adaptation, one may expect to model the opponent’s current policy $\pi^t_2$ by **extrapolating** from past interactions with $\pi^{1:t-1}_2$, even though $\pi^t_2$ may differ from $\pi^{1:t-1}_2$. This would require challenging sequential modeling. The key insight of sFP is that, instead of predicting $\pi^t_2$, we only need to model the average of $\pi^{1:t-1}_2$, thus **interpolating** among historical interactions. This also helps justify why in-context learning is both a principled and practical implementation choice, as we explain next.
>
> (2). sFP also justifies why (smoothly) best responding to such a hypothetical model is sufficient for provable performance, motivating our BoN module.
>
> ## Response to W2, Q1, and Q2:
> - In-context learning (ICL) v.s. Fine-tuning: (1). Computation latency and overhead: For online adaptation at deployment time, we may not have access to gradient-based parameter updates, especially in black-box settings. Moreover, such settings often have strict latency requirements. In contrast, ICL only requires LLM inference and thus incurs much lower latency and computational overhead. (2). Sample efficiency: this is the main reason we did not use fine-tuning even in our experiments. For fine-tuning to be effective, it must first accumulate sufficient interaction data with the opponent. In contrast, ICL is known to work well in zero-shot/few-shot settings, leading to much lower burn-in cost. (3). Theoretical soundness of ICL: Prompt-based ICL is not merely a heuristic choice; it has been shown to be closely connected to Bayesian inference, which is a reasonable estimator in our setting since clear latent variables exist, such as the opponent’s private information or constraints. It can also connect to estimators such as MLE via the Bernstein–von Mises theorem. More importantly, under certain Gaussian data assumptions, ICL can provably implement the empirical averaging mechanism used in the proof of Theorem 4.1 of Park et al. (2025). Therefore, we believe using ICL for opponent modeling is a theoretically principled design choice.
>
> - Empirical opponent error analysis: Although directly comparing against fine-tuned opponent models is challenging, in our controlled setting we can compare against the oracle opponent itself. We kindly refer the reviewer to Figures 10 and 11, where opponent modeling achieves increasingly better evaluation accuracy.
>
> - Sensitivity to the opponent model quality: We additionally run linear regression on the data pairs $\{(x_i, y_i)\}$, where for each episode $i$, $x_i$ is the opponent model accuracy and $y_i$ is the reward. We obtain an $R^2$ of 0.71, suggesting that performance is well explained linearly by opponent model quality.
>
> ## Response to W3:
> - For the computation and performance analysis, we kindly refer the reviewer to Table 2, where we vary the main scaling factor $N$ and report the corresponding computation tokens and performance gains. The results show that scaling $N$ incurs reasonable cost while yielding considerable gains.
> - We additionally plot the computation-performance scaling behavior of our approach against naively scaling larger models. The results can be found in Figure 1 of [result](https://drive.google.com/file/d/1jRJqvTt8RZaSLi54gnBAANBZuo4avCZB/view?usp=sharing), where we see that our approach makes more effective use of the compute budget.
>
> ## Response to W4 and Q4:
> Firstly, we believe negotiation is a well-recognized benchmark for strategic reasoning (Lewis et al., 2017; Davidson et al., 2024; Bianchi et al., 2024). To further evaluate our approach, we additionally include two domains: repeated colonial blotto and repeated liar’s dice. The former stress-tests the exponentially large action space, while the latter stress-tests settings involving strategic deception and bluffing. Both are core challenges for strategic reasoning. We refer the reviewer to the results in Table 1 of [result](https://drive.google.com/file/d/1jRJqvTt8RZaSLi54gnBAANBZuo4avCZB/view?usp=sharing).

---

> > ### Author Rebuttal · Reviewer_z1u3 · 2026-04-02
> >
> > Thank you for the detailed rebuttal. I appreciate the authors’ careful clarifications on the theory–practice gap, the motivation for using in-context opponent modeling instead of fine-tuning, and the additional discussion on compute scaling and broader domains. Overall, the response strengthens my confidence in the paper’s practical and technical positioning, and I am happy to maintain my current score.
> >
> > That said, my concern about the faithfulness of the prompt-based opponent model to the theoretical assumptions is only partially resolved. While the rebuttal provides reasonable motivation and some indirect supporting evidence, I still think this aspect could be analyzed more directly.

---

> > > ### Author Response · Authors · 2026-04-06
> > >
> > > We thank the reviewer for acknowledging our rebuttal efforts, and we are glad to hear that many of the concerns have been addressed. Below, we further respond to Limitation 1 with a more direct analysis.
> > >
> > > **Further discussion of theoretical assumptions.** For the action space, the bound in our Theorem 4.4 depends only on the total variation distance, rather than on the size of the action space. For Proposition 4.2, the dependence on the action space is only logarithmic. Therefore, the main requirement on the action space is finiteness, rather than a moderate size.
> > >
> > > **Direct empirical analysis:** We also directly assess whether the prompt-based opponent model is faithful to the bounded-total-variation-distance assumption by measuring the total variation distance between our opponent model and the actual opponent. We refer the reviewer to the detailed results and discussion in [NEW RESULT](https://drive.google.com/file/d/16hv1QpQa2TPR21ehGp2w3bPb3MLDT6Hw/view?usp=sharing).
> > >
> > > Finally, we note that upper-bounding total variation distance is a very common assumption across various areas of the RL literature, including online RL [1], offline RL [2], non-stationary RL [3], and robust RL [4]. In many cases, it can serve as a minimal assumption for deriving theory, and it also connects naturally to other notions of probability divergence.
> > >
> > > ---
> > >
> > > [1] Subramanian, Jayakumar, et al. "Approximate information state for approximate planning and reinforcement learning in partially observed systems." *Journal of Machine Learning Research* 23.12 (2022): 1-83.
> > >
> > > [2] Foster, Dylan J., et al. "Offline Reinforcement Learning: Fundamental Barriers for Value Function Approximation." *Conference on Learning Theory*. PMLR, 2022.
> > >
> > > [3] Cheung, Wang Chi, David Simchi-Levi, and Ruihao Zhu. "Reinforcement learning for non-stationary Markov decision processes: The blessing of (more) optimism." *International Conference on Machine Learning*. PMLR, 2020.
> > >
> > > [4] Panaganti, Kishan, and Dileep Kalathil. "Sample complexity of robust reinforcement learning with a generative model." *International Conference on Artificial Intelligence and Statistics*. PMLR, 2022.

---

### Official Review · Reviewer_zBZj · 2026-03-12

**Soundness:** 2
**Presentation:** 2
**Significance:** 3
**Originality:** 2
**Overall Recommendation:** 4
**Confidence:** 4

**Summary:**

This paper proposes a two-stage inference-time framework for LLM negotiation: (1) belief formation via in-context opponent modeling from interaction history, and (2) best response selection via Best-of-N (BoN) sampling with opponent simulation. The approach is framed through Smooth Fictitious Play (sFP) theory, with a claimed O(√T) no-regret guarantee. Experiments on buyer-seller and resource exchange games across Claude-Sonnet-4, Qwen3, and Llama-3.3 show improvements over direct prompting and some adaptive baselines.

**Compliance With Llm Reviewing Policy:**

Affirmed.

**Final Justification:**

In the rebuttal, the authors added new baseline comparisons, computed-efficiency analysis against larger models, and the empirical validation of Theorem 4.4 addresses my main concerns.

New results in the attached PDF should be added to the preprint in the camera-ready version, given acceptance.

I am satisfied that the core concerns have been resolved.

**Key Questions For Authors:**

1. How does BoN-oppo-simulation compare against Strategist [1] and RL-based dialogue agents [4] under the same game settings? Without external baselines, the contribution cannot be properly assessed.

2. If you normalize for total inference compute (N candidates × simulation depth × both players), how does BoN-oppo compare to simply using a larger/better model with the same compute budget?

3. Can you independently measure opponent modeling accuracy and validate Theorem 4.4 empirically?

**Limitations:**

The paper's primary limitations — many of which were raised in the prior ICLR 2026 review cycle — remain insufficiently addressed: (1) missing external baselines (Strategist [1], K-Level Reasoning [2]) make the evaluation insular, (2) the blurred line between prompt engineering and genuine inference-time scaling undermines the novelty claim, (3) compute overhead is unquantified, (4) theoretical guarantees require perfect opponent modeling which is not achieved. The presentation issues (vspace abuse, typos) further weaken the submission.

**Strengths And Weaknesses:**

### Strengths

1. **Theoretical framing**: Connecting LLM negotiation to Smooth Fictitious Play is intellectually interesting. Theorem 4.4 (optimality gap scales linearly with opponent modeling error) provides a useful conceptual framework, even if the assumptions are strong.

2. **Training-free and model-agnostic**: The method requires no fine-tuning and is demonstrated across three LLM families (Claude, Qwen, Llama), making it immediately deployable.

3. **Evaluation against adaptive baselines**: Testing against specialized strategies (AI feedback, selective experience reflection, private information prediction) rather than only naive baselines is commendable.

### Weaknesses

1. **Limited novelty and missing critical baselines**: The self-improvement via opponent modeling and search paradigm has been explored by Strategist [1], which combines LLMs with MCTS for iterative strategy refinement in competitive games, K-Level Reasoning [2] for strategic decision-making, and ShapeLLM [3] for opponent shaping in multi-agent LLM interactions. On the RL side, goal-conditioned value functions for multi-turn dialogue planning [4] and offline RL with hindsight regenerations for interactive dialogue agents [5] are also directly relevant but uncited. The paper does not compare against these baselines. This was a key criticism in the prior submission at ICLR 2026, where the paper was rejected with 3 negative reviewers specifically raising the concern that empirical claims feel insular — tested only against in-house or weak baselines. The distinction between this work and prompt engineering (the framework relies heavily on specialized prompts for brainstorming, reflection, and opponent summarization in Appendix A) is also unclear and overstated.

2. **No compute-efficiency analysis**: BoN with N=8 and multi-step opponent simulation requires potentially ~48x the compute of a single response. The paper never addresses whether this is better than simply using a larger/better model with the same compute budget. Without compute-normalized comparisons, the improvements may simply reflect throwing more compute at the problem.

3. **Writing quality and presentation**: The paper is difficult to read due to a combination of excessive `\vspace` compression and disjointed writing flow. The main body is overly dense — several proofs and derivations should be moved to the appendix to give core ideas room to breathe. Multiple typos further indicate insufficient proofreading: Figure 1 reads "sreller starts second" instead of "seller starts second". The manuscript needs to be proofread and formatted more carefully.

[1] Light, J., Cai, M., Chen, W., Wang, G., Chen, X., Cheng, W., Yue, Y., & Hu, Z. "Strategist: Self-improvement of LLM Decision Making via Bi-Level Tree Search." arXiv:2408.10635, 2024. https://arxiv.org/abs/2408.10635

[2] Zhang, Y., Mao, S., Ge, T., Wang, X., Xia, Y., Lan, M., & Wei, F. "K-Level Reasoning: Establishing Higher Order Beliefs in Large Language Models for Strategic Reasoning." arXiv:2402.01521, 2024. https://arxiv.org/abs/2402.01521

[3] García Segura, M. E., Hailes, S., & Musolesi, M. "Opponent Shaping in LLM Agents." ICLR 2026. https://openreview.net/forum?id=yJoHTqUNry

[4] Hong, J., Dragan, A., & Levine, S. "Planning without Search: Refining Frontier LLMs with Offline Goal-Conditioned RL." NeurIPS 2025.

[5] Hong, J., Lin, J., Dragan, A., & Levine, S. "Interactive Dialogue Agents via Reinforcement Learning on Hindsight Regenerations." arXiv:2411.05194, 2024. https://arxiv.org/abs/2411.05194

---

> ### Author Rebuttal · Authors · 2026-03-31
>
> We thank Reviewer zBZj for the detailed feedback. We believe there may be misunderstandings regarding what we have revised during the last ICLR 2026 cycle, since before the incident occurred and reviews were locked, we received no feedback from 3 of the 4 reviewers. As a result, we believe some of the initial negative reviews from that cycle do not fairly reflect the quality of our paper. Since then, we have also substantially revised the paper in many aspects with most limitations listed by the reviewer already properly addressed.
>
> ## Response to W1 & Q1:
>
> - Regarding novelty: we kindly remind the reviewer that **none of the cited works considers using an opponent model**. Opponent shaping in [3] refers to the distinct problem of steering the opponent toward certain behaviors or equilibria. In contrast, opponent modeling is a technique to infer the behavior of an unknown opponent from interaction data. Moreover, [3] uses a model-free approach rather than opponent modeling. For [1, 2, 4, 5], they do not consider online interaction with an actual unknown opponent either, so no opponent model can be built; in addition, [4, 5] study a single-agent MDP setting. Meanwhile, our second BoN module also involves non-trivial design considerations beyond the standard MCTS template, as clarified in our response to Reviewer U8av.
> - Regarding missing baselines: since the ICLR 2026 rebuttal, **we have already added three new sets of baselines relevant to our setting** and are now further extending two comparisons requested by the reviewer. We did not compare with [1, 4] originally because they do not solve our online adaptation problem: their approaches are not designed to leverage incremental online interactive feedback and continuously update the policy against unknown dynamic opponents. Still, we are happy to extend them as much as possible to our setting and ***refer the new experiments and results to*** [pdf](https://drive.google.com/file/d/1L_Us7gfLQrVVX7gfcR30qxSexx9EMKxX/view).
> - Regarding the difference between prompt engineering and our work: the distinction lies in whether the method can systematically exploit additional inference-time computation, specifically through scaling at the output level. **This input-level v.s. output-level categorization is not our own invention, but follows the cornerstone framing of Snell et al. (2024).** Of course, prompt design is still necessary for all inference-time techniques and we never claimed to avoid it. In our revised version, we have made this distinction clearer, by using the specific nomenclature "inference-time scaling" rather than referring to our work only as the generic inference-time techniques.
>
> ## Response to W2 & Q2:
> - We kindly remind the reviewer that **we do have compute-efficiency analysis in Table 2**. Our default choice is $N=5$, which requires only 12x~25x compute rather than 48x, compared with no thinking, i.e., no inference-time scaling at all. This is quite reasonable compared with prior work, where longer thinking or BoN often incurs substantially larger compute overhead (Snell et al., 2024; Brown et al., 2024; Muennighoff et al., 2025).
> - We fully agree that understanding how to use test-time computation efficiently is a central question, and this is exactly one of the main goals of our experiments. This is why we compare many different strategies for scaling inference-time compute, including BoN variants and our new self-simulation baseline, all under the same key scaling factor $N$.
> 	- ***New experiments: as suggested by the reviewer, we compare the scaling behaviors of our approach and larger models and refer the results to Figure 1 of*** [pdf](https://drive.google.com/file/d/1L_Us7gfLQrVVX7gfcR30qxSexx9EMKxX/view)
>
>
> ## Response to W3:
> We believe we already deferred all proofs to the appendix, but we still genuinely thank the reviewer for raising this point and will move some motivating discussions there as well for improved readability.
>
> ## Response to Q3:
>
> For opponent modeling accuracy, we kindly refer the reviewer to the existing Figures 10 and 11, where opponent modeling achieves the best and most stable evaluation accuracy compared with other evaluation approaches.
>
> To validate Theorem 4.4, we run linear regression on data pairs ${(x_i, y_i)}$, where for each episode $i$, $x_i$ is the opponent model accuracy and $y_i$ is the reward. We obtain an $R^2$ of $0.71$, implying that better performance is well explained linearly by better opponent model quality, consistent with the prediction of Theorem 4.4.
>
> ***Finally, we thank the reviewer for pointing out all the related works and will make sure to include proper discussions with them in our paper.***

---

> > ### Author Rebuttal · Reviewer_zBZj · 2026-04-01
> >
> > Thank you for the thorough rebuttal. The new baseline comparisons, compute-efficiency analysis against larger models, and the empirical validation of Theorem 4.4 address my main concerns.
> >
> > New results in the attached PDF should be added to the preprint in the camera-ready version, given acceptance.
> >
> > I am satisfied that the core concerns have been resolved.

---

### Official Review · Reviewer_rBaC · 2026-03-13

**Soundness:** 3
**Presentation:** 3
**Significance:** 3
**Originality:** 3
**Overall Recommendation:** 5
**Confidence:** 3

**Summary:**

The paper proposes an LLM inference-time strategy for repeated play in negotiation games, where at every round, two agents repeatedly interact to reach a deal and receive some reward based on the outcome. The proposed strategy is inspired by fictitious play/no-regret learning; at every round, an agent simulates their opponent based on history and “best responds” by choosing the best out of N generated strategies. The authors evaluates their method on two games, showing that it outperforms various baseline methods and existing inference-time strategies.

**Compliance With Llm Reviewing Policy:**

Affirmed.

**Final Justification:**

The rebuttal addressed my main concern.

**Key Questions For Authors:**

See weaknesses section above. Also, I found it interesting that, compared to other methods, the proposed method achieves more performance gains in the zero-sum game than the cooperative game. Is this behavior expected/can the authors comment on this further?

**Limitations:**

Yes

**Strengths And Weaknesses:**

Strengths:
- The proposed method for online adaptation is well-grounded in standard no-regret learning procedures. The paper addresses challenges of implementing such procedures for language models, proposing a nice solution that is fairly lightweight. The method is general and could be applied to strategic settings beyond negotiation games. The experiments are extensive and compelling, especially for the zero-sum game.

Weaknesses:
- In the experimental evaluations, not much is said about the opponent strategy. In particular, how adaptive is the opponent? If the opponent is fairly static, the opponent belief formation and best response steps should be easy. I think the more interesting case is if the opponent is also adaptive in a meaningful way; the agent must learn something about how the opponent’s learning process. This is also where the value of fictitious play/no-regret learning becomes more apparent; if the opponent is static, these algorithms would be unnecessary. Although the paper does evaluate performance against an explicitly adaptive opponent, it does not include comparisons with baselines or other methods in this setting.
- It is hard to isolate what is affecting reward over time, e.g. better opponent modeling over time, better BoN strategization, degrading opponent performance. It would be interesting to e.g. compare the performance gains of using only one of the two steps vs using both steps.

---

> ### Author Rebuttal · Authors · 2026-03-31
>
> We thank Reviewer rBaC for the positive feedback. We are encouraged that the reviewer finds our method well-grounded and experiments extensive and compelling. We here address the reviewer’s concerns as follows.
>
> ## Response to W1:
>
> - Regarding the existing opponents: apart from the default opponents that naturally make adaptive decision-making through cumulative history conditioning, we have explicitly examined four kinds of adaptive opponents: (1) the opponents also using our approach (Fig. 4b, Fig. 10). We regard this as a very strong adaptive opponent since it has exploited other baselines very well as demonstrated across our experiments. (2). Fu et al., 2023 (3) Xu et al., 2023 (4) Yu et al., 2025, which are representative works from literature and designed to be adaptive in very meaningful ways, e.g., self-refining, self-reflecting, and inferring private information, etc. Now we include two new experiments suggested by the reviewer as follows and refer the results to [result](https://drive.google.com/file/d/1CYuLmkfO6V5-HqdVv5K-kwFdry_0hogr/view?usp=sharing).
> - New experiment 1: As suggested by the reviewer, we now also include baselines against these three opponents (2), (3), (4), which also help us understand how strong these opponents are. We can see these three external adaptive methods are strong, where some BoN baselines only achieve less than 0.5 reward against them (since the evaluation is symmetrized and the reward is normalized, equally strong methods should achieve 0.5 reward), although still underperform our approach.
> - New experiment 2: In response to the reviewer’s suggestion, we add two more adaptive opponents: (a). Meta-adaptive opponent: adaptively choosing among the strongest 3 meta prompt strategies from Sec. 4 using an adversarial bandit subroutine (b). Adversarial probing: we prompt the opponent to start with a fair price and incrementally demand more profit (i.e., increase or decrease its price) every episode. However, it should follow a conditional stabilization rule: if its proposal is rejected, it interprets this as a hard boundary and 'holds firm' at the previous successful deal price for the next three episodes before attempting to escalate again. One can expect a smart agent should learn this behavior and strategically reject it at some times. Again for ease of presentation, we symmetrize the game and normalize the rewards, where our approach still achieves the best performance.
>
> ## Response to W2:
> - Isolating the effects of better opponent modeling: We kindly refer the reviewer to our existing results, Figure 10&11, where opponent modeling provides better and better accuracy across time. Further, we run linear regression on the data pairs {(x_i, y_i)}, where for each episode i, x_i is the opponent model accuracy and y_i is the reward. We get r2 of 0.71, implying that better performance can be very well explained by better opponent model quality.
>
> - Isolating the effects of better BoN strategizing: This is exactly why we have considered the BoN-eval method whether we keep BoN but remove opponent modeling/simulation.
>
> - Isolating degrading opponent performance: Quantitatively, we didn’t observe any unreasonable behavior and find they still follow similar reasoning and response patterns. Example outputs can be found in https://github.com/llmnegotiationsubmission/llmnegotiationsubmission. Besides, to further corroborate this point, we have also seen our approach being still effective against strong adaptive opponents above with the ability to improve instead of degrading.
>
> Meanwhile, it is a bit unclear how to remove the BoN step but only keep the opponent modeling step since the opponent model is built for BoN in a sparse reward environment. That being said, the comparison with turning on the thinking mode of the base model also highlights the effectiveness of the BoN step as a kind of inference-time scaling strategy compared with just longer CoT.
>
>
> ## Response to Q1:
>
> We thank the reviewer for the insightful comments. We first kindly remind the reviewer that the buyer-seller games are not zero-sum/constant-sum since when no deals are made, no agents get any rewards, which could enable interesting strategic behaviors like deliberative rejection. Similarly, the resource exchange game is also a mix of cooperation and competition. Now to answer the reviewer's question, we believe the reason why we see more performance gains in the zero-sum game than the cooperative game is related to LLM behavioral priors. Most modern LLMs undergo safety and alignment training that biases them toward "helpfulness" and "fairness", which happen to already be a strong strategy, despite that they are explicitly asked to maximize their own reward.

---

> > ### Author Rebuttal · Reviewer_rBaC · 2026-04-03
> >
> > Thank you for the response! The new experiments seem convincing to me and address my main concern. I've updated my score.

---

### Official Review · Reviewer_U8av · 2026-03-14

**Soundness:** 2
**Presentation:** 3
**Significance:** 1
**Originality:** 1
**Overall Recommendation:** 3
**Confidence:** 3

**Summary:**

Large Language Models have demonstrated great promise in various decision-making tasks which involve single agent and stationary environments. However, the behavior of repeated and strategic interactions with dynamic/unknown opponents is under exploration. In particular, LLMs often fall short in adapting online based on interaction feedback. To address the challenge, the paper proposes BoN-oppo-simulation, which (1) introduces an auxiliary opponent model to imitate the average behavior of the opponent for belief formation; (2) deployed a best-of-N sampling strategy by simulating against the opponent model for best response. The empirical results show that bon-oppo-simulation demonstrates significant improvement on two distinct forms of repeated negotiation games when compared to various baselines.

**Compliance With Llm Reviewing Policy:**

Affirmed.

**Final Justification:**

The additional clarifications and new evidences have addressed most of my concerns, particularly around the framing of sFP, robustness to imperfect opponent modeling, and the distinction from alternative prompt-based designs. Therefore, I raised my score. However, I still lean somewhat toward reject, mainly because I remain unsure whether the overall contribution is sufficiently novel and broad in scope relative to existing planning/simulation-based approaches.

**Key Questions For Authors:**

1. What is the length (in tokens or turns) of each iteration in the experiments? How does the performance change as the information gets complex, and how good a model is at predicting very complex opponent action? For example, what happens when the context length approaches the base model’s training budget. Prior work has shown that the model's instruction following capability can drop near the token limit because the training distribution often has a long tail.

2. How does the model perform against different opponents, such as running Llama agents against GPT models?

3. How does the method perform when the agent does not have full access to information about the opponent?

**Limitations:**

Yes

**Strengths And Weaknesses:**

**Strengths**

1. The problem on how to enable LLMs adaptation capability during test-time is interesting, and it is indeed a core question to study for enabling the model's self-evolving and continuous improvement capability.

2. The connection between the method and smooth Fictitious Play in classical game theory is novel.

**Weaknesses**

1. Although the paper frames step 1 as belief formation inspired by fictitious play, the actual implementation is just to prompt an LLM to summarize opponent behavior and predict future responses, the game-theoretic framing is a bit metaphorical.

2. Following on 1, the success of the algorithm depends heavily on whether the simulated opponent can represent the real opponent. However, the simulated opponent is just another LLM prompt, and there is no method guarantee calibration or likelihood accuracy. The effectiveness may depend highly on tasks and models’ capability. It is not intuitive to me how the method can be used with a slightly weaker model, and in an environment where the opponent info is only partially observable, which is the case in many real-world settings. The results show good improvement in this paper can be explained by the opponent here is the base LLM itself, which avoids the case that the model cannot predict the opponent well.

3. The method itself, from an implementation perspective, is not novel. It resembles Monte-Carlo lookahead planning, which starts with sampling a candidate action, rolling out the future, and then picking the action that leads to the best future. This has existed in literature like Monte Carlo Tree search, model-based RL, and the method itself doesn’t seem to be a fundamentally new algorithm.

4. For both buyer-seller negotiation, and resource exchange game, the tasks are repeated negotiation with only a short horizon and structured, which make it a bit narrow to claim general strategic reasoning.

---

> ### Author Rebuttal · Authors · 2026-03-31
>
> We thank Reviewer U8av for the detailed feedback. We believe there are some misunderstandings regarding our problem setting and experimental setups.
>
> ## Response to W1 & W3:
>
> Although exactly implementing sFP in language-based negotiation is intractable, we respectfully disagree that game-theoretic framing is merely metaphorical. We also note several non-trivial considerations compared to classical MCTS.
>
> *   **sFP provides the theoretical principle for why we can rely on opponent modeling instead of sequential modeling in a task where the opponent is evolving.** For online adaptation where the opponent can evolve, one may expect to model the opponent $\pi^t\_2$ by ***extrapolating*** from existing interactions with policies $\pi^{1:t-1}\_2$, where $\pi^t\_2$ can differ from $\pi^{1:t-1}\_2$. This seems to necessitate challenging sequential modeling and is beyond the capabilities of standard opponent modeling. However, the theoretical elegance of sFP is that instead of predicting $\pi^t_2$, we only model the average of $\pi^{1:t-1}$, thus only ***interpolating*** among historical interactions/policies. Therefore, the usage of an opponent modeling module and its specific prompt design are not just heuristic choices. Of course, if the opponent model happens to also approximate $\pi_2^t$ well, we can expect stronger results (e.g., Theorem 4.4 and Prop 4.2 by strengthening regret to dynamic regret).
> *   **Following the first, prompting and context conditioning is exactly implementing such interpolation instead of extrapolation via in-context learning.** As shown in our prompts (Appendix), we never explicitly ask the opponent model to analyze the trend of the opponent or extrapolate future behaviors, but only to simulate given the history. Furthermore, in-context learning is proven to be a principled statistical estimator (see references [a, b]).
> *   **Mapping classical principles into LLM primitives is an established paradigm.** For example, *Reflexion* and Xia et al. (2024a) implement the classical idea of actor-critic and dueling bandits' principles via LLM. Although BoN shares conceptual similarities with MCTS, many of our design choices—e.g., how to strategically explore text spaces and how to leverage an opponent model for evaluation in an evolving environment—are novel to the traditional MCTS template.
>
> ## Response to W2 & Q3:
>
> We believe there are some misunderstandings regarding our experimental setups.
> *   **Base Model Mismatch:** The opponent is ***never*** the base LLM itself, also with always a mismatch with the opponent model's LLM **to avoid self-modeling bias**. For example, when the real opponent is powered by Gemini-Flash, the opponent model ranges from its weaker version (Gemini-Flash-Lite) to Qwen, Llama, and Claude, and vice versa (see Table 3, as referenced in the main paper). This strictly covers cases where the opponent model is both stronger and weaker than the true opponent.
> *   **Partial Observability:** We **never** have full access to the opponent's information; our setting is strictly partially observable. Only the public negotiation history is accessible to the agent, while critical information like the opponent’s budget, constraints, prompt, and internal thinking **remain strictly private**.
>
> ## Response to W4 & Q1:
>
> *   **Task complexity:** While there are higher-dimensional domains than negotiation, it is a well-recognized benchmark for strategic reasoning (Lewis et al., 2017; Davidson et al., 2024; Bianchi et al., 2024), not to mention we extend this to the more complex repeated setting. Following these, the number of turns in each episode is flexible and determined most by the agents themselves (e.g., an episode stops when both agents reach an agreement). Finally, the action space of the agents allow unstructured and free-format communication.
> *   **Context limits:** We view context management as a very interesting but complementary technical hurdle. Our sFP-guided opponent model also facilitates memory management since its goal is to approximate the average effects of historical policies rather than performing exact information retrieval (i.e., looking for a needle in a haystack).
>     *   ***New Experiment:*** To validate this hypothesis, we ran an ablation dropping each historical episode from the context with probability $p$, which is unbiased for modeling the average, but can hurt information retrieval. At $p=0.3$, performance decreases by only 12%; at $ p=0.6 $, performance drops by just 23%.
>
> ## Response to Q2:
>
> We kindly direct the reviewer to **Table 3**, where we already studied opponent generalization across different model architectures.
>
> We thank the reviewer for their engagement and happy to take any new feedbacks.
>
> ---
> [a]. An Explanation of In-context Learning as Implicit Bayesian Inference, ICLR 2022
>
> [b]. Supervised Pretraining Can Learn In-Context Reinforcement Learning, NeurIPS 2023

---

> > ### Author Rebuttal · Reviewer_U8av · 2026-04-03
> >
> > Thank you for the detailed rebuttal. I appreciate the effort to address the concerns, but I still have a couple of reservations:
> >
> > > sFP connection
> >
> > Thanks for the additional intuition for the connection to sFP, which, however, remains largely conceptual. I do not see empirical evidence that the proposed method behaves differently from a generic prompt-based opponent model, nor that the sFP perspective leads to measurable advantages over alternative formulations.
> >
> > > opponent modeling reliability
> >
> > Sorry for the confusion in the previous response, the core concern is not about whether the opponent model is identical to the opponent, but rather how sensitive the method is to inaccuracies in the opponent model, which I don’t think is fully addressed. In particular, I think adding (1) analysis of robustness to poor opponent modeling, (2) calibration or likelihood quality of the simulated opponent, or (3) failure cases when the opponent behavior is harder to predict, would largely resolve the concern on whether the success rely heavily on a well-behaved or predictable opponent.
> >
> > > novelty vs existing planning approaches
> >
> > It is true that a lot of works is “mapping classical principles into LLM primitives”, but it would be helpful to also articulate what is fundamentally new beyond adapting existing planning ideas to language model setting.

---

> > > ### Author Response · Authors · 2026-04-05
> > >
> > > We are glad that some of the reviewer’s original concerns have been addressed. We also thank the reviewer for further clarifying these questions, which allows us to respond more directly below:
> > >
> > > # R1: Exploring Alternatives
> > >
> > > Since the literature on opponent modeling for LLMs is still limited, we are not aware of a mature generic formulation for this setting. Following the reviewer’s concerns:
> > >
> > > - **Difference with alternatives:** Existing LLM opponent-modeling approaches can be roughly grouped into two types:
> > >     - Type I: theory-of-mind-style methods that only infer private information or hidden reasoning of a hypothetical opponent.
> > >     - Type II: minimax-style methods in which the opponent model is designed as a strong adversary for worst-case evaluation.
> > >
> > >     By contrast, an opponent model for simulation is not explicitly built, as  noted by the very recent work of Yu et al. (2025). Our sFP-style opponent model aims to interpolate the opponent’s historical policies. This leads to design choices such as summarization, role-playing, and consistency with history, rather than private-information inference or adversarial modeling. While this may appear natural, or even generic, from the viewpoint of traditional multi-agent RL, it is not a generic choice in the current LLM literature described above. Besides, all these implementations are prompt-based.
> > >
> > > - ***New Empirical Evidence 1: comparing alternatives*** Following the reviewer’s suggestion, we compare against four alternatives: **(1)** Opponent model (generic), **(2)** Opponent model (extrapolative), **(3)** Opponent model (Type I above: private-information prediction), and **(4)** Opponent model (Type II above: adversarial). We refer to the concrete details of these natural alternatives and the result table in [NEW EXP 1](https://drive.google.com/file/d/1KH-ZMkqgSHddIG6sje1Wj6vi6BusAtot/view?usp=sharing). The results show that the sFP perspective, though simple, leads to a more effective opponent-model design than these alternatives. For more alternatives such as value prediction, we also refer to our response to Reviewer zBZj.
> > >
> > > # R2: Robustness to Opponent Model Accuracy
> > >
> > > Following the reviewer’s suggestion, we explicitly test robustness against the following deliberately designed opponents.
> > >
> > > - Meta-adaptive opponent. Arising from the discussion with Reviewer rBaC, the opponent adaptively selects among the five strongest meta-prompt strategies from Sec. 4 using an adversarial bandit subroutine. The randomization of this principled adversarial bandit algorithm make prediction difficult.
> > >
> > > - Opponent with oracle access to misalign with the simulation. To construct a more explicitly hard-to-predict opponent, before each episode, the opponent obtains oracle access to the trajectories simulated by the opponent model. It then enumerates different brainstormed strategies and interacts with the actual decision-maker to choose the one that maximally mismatches the simulated trajectories before committing to it in the real interaction.
> > >
> > > ***New Empirical Evidence 2:*** We report both performance and calibration metrics, the TV distance between the opponent model and the actual opponent, in this [NEW EXP 2](https://drive.google.com/file/d/1kRwEHzOkwbrHZ0oQwq_rNSw5VKTX4Grg/view?usp=sharing). The results show that our method does not rely on highly accurate prediction of a fast-changing opponent in order to outperform other baselines, which further supports the theoretical appeal of the sFP principle.
> > >
> > > # R3: Novelty and Main Focus
> > >
> > > We agree that operations such as branching and rollout are not new. However, our goal is not to solve planning, but to enable online adaptation, and the connection between the two is not obvious. The novelty lies in **reducing the more challenging problem of online adaptation to two decoupled, subproblems**, rather than in proposing a fundamentally new planning algorithm. Still, adapting BoN effectively requires new efforts, including identifying best practices for candidate evaluation for LLM, understanding inference-time scaling behaviors, and enabling structured strategic brainstorming to better explore the semantic space. Besides, planning does not address the problems of whether and how we can build a simulator against an *online* opponent.
> > >
> > > Finally, for our purpose as stated in our title and main question, we feel it is better to adapt a planning-style BoN method to instantiate the best-response module, as this helps ***isolate and understand*** the best practices behind opponent modeling, using simulation or not, clarify the inference-time scaling behavior, etc. We do acknowledge that we did not invent a fundamentally new planning algorithm and will note this as a future direction, since improved planning may be used to further strengthen the overall framework.
> > >
> > > ---
> > >
> > > Again, we truly appreciate the reviewer for taking the time to point out additional empirical evidence needed. We hope the new evidence address the reviewer’s concern.

---

### Decision · Program_Chairs · 2026-04-30

**Decision:**

Accept (regular)

**Comment:**

Despite an initial mixed reaction to the paper, the rebuttal and discussion has moved the reviewers toward a more positive opinion of the paper. As things currently stand, the paper has good support from two reviewers, and weak support from one. Reviewer U8av voted for weak rejection.

While Reviewer U8av was not able to participate in the discussion with the author, I have reached out to them privately after the discussion phase to get a sense for whether they would support acceptance despite their weak rejection recommendation. In our discussion, the reviewer mentioned the following:

> The authors’ latest response addressed several of my main concerns, particularly through the additional experiments on alternative formulations and robustness, which is why I raised my score in the final justification from 2 to 3.
>
>  I still retain some reservations about novelty. More specifically, I am not yet fully convinced that the core technical contribution goes substantially beyond combining a prompt-based opponent model with simulation-based best-of-N response selection. In particular, while the smooth fictitious play perspective is an interesting lens, it remains somewhat unclear to me to what extent it leads to a fundamentally new algorithmic ingredient, rather than a well-motivated framing and a particular prompt/design choice within a broader family of planning- and simulation-based methods. I also remain somewhat unsure about how broadly the contribution extends beyond the repeated negotiation settings studied here.
>
>  That said, I would be comfortable with the paper being accepted, especially in light of the generally positive assessments from the other reviewers. For me personally, however, I still feel that a score of 3 is the right assessment for the paper.

Taking into account the comments of all reviewers, and the general dynamic of the discussion, I support acceptance. I warmly encourage the authors to keep into account the comments of all reviewers when preparing their next version.